



# Contributions of nitrated aromatic compounds to the light absorption of water-soluble and particulate brown carbon in different atmospheric environments in Germany and China

Monique Teich[1], Dominik van Pinxteren[1], Michael Wang[2], Simonas Kecorius[1], Zhibin Wang[1,5],

Thomas Müller[1], Griša Močnik[3,4], Hartmut Herrmann[1]

[1]Leibniz Institute for Tropospheric Research, TROPOS, 04315 Leipzig, Germany

[2]McMaster University, Hamilton, ON L8S 4L8, Canada

[3]Aerosol d.o.o., 1000 Ljubljana, Slovenia

[4]Condensed Physics Department, Jožef Stefan Institute, Ljubljana, Slovenia

[5]Now at: Multiphase Chemistry Department, Max Planck Institute for Chemistry, 55128 Mainz, Germany

Correspondence to: Hartmut Herrmann (hartmut.herrmann@tropos.de)

**Abstract.** The relative contribution of eight nitrated aromatic compounds (NACs, nitrophenols + nitrated salicylic acids) to the light absorption of aqueous particle extracts and particulate brown carbon were determined from aerosol particle samples

collected in Germany and China.

High-volume filter samples were collected during six campaigns, performed at five locations in two seasons: (I) two campaigns with strong influence of biomass burning (BB) aerosol – at the TROPOS institute (winter, 2014, urban background, Leipzig, Germany) and the Melpitz research site (winter, 2014, rural background); (II) two campaigns with strong influence from biogenic emissions – at Melpitz (summer, 2014) and the forest site Waldstein (summer, 2014, Fichtelgebirge, Germany), and

(III) two CAREBeijing-NCP campaigns – at Xianghe (summer, 2013, anthropogenic polluted background) and Wangdu (summer, 2014, anthropogenic polluted background with a distinct BB-episode), both in the North China Plain.

The filter samples were analyzed for NAC concentrations and the light absorption of aqueous filter extracts was determined. Light absorption properties of particulate brown carbon were derived from a seven-wavelength Aethalometer during the campaigns at TROPOS (winter) and Waldstein (summer). The light absorption of the aqueous filter extracts was found to be

pH dependent: at pH 10, the aqueous light absorption coefficient $Abs_{370}$ and the mass absorption efficiency ($MAE_{370}$) at 370 nm were a factor of 1.6 and 1.4 larger than at pH 2, respectively. In general, $Abs_{370}$ ranged from 0.21–21.8 Mm$^{-1}$ under acidic conditions and 0.63–27.2 Mm$^{-1}$ under alkaline conditions, over all campaigns. The observed $MAE_{370}$ was in a range of 0.10– 1.79 m$^2$ g$^{-1}$ and 0.24–2.57 m$^2$ g$^{-1}$ for acidic and alkaline conditions, respectively. For $MAE_{370}$ and $Abs_{370}$, the observed values were higher in winter than in summer, in agreement with other studies. Furthermore, it was found that the $MAE_{370}$ values in

winter in Germany exceeded those of the Chinese summer background stations (average of 0.85±0.24 m$^2$ g$^{-1}$ compared to 0.47±0.15 m$^2$ g$^{-1}$). The lowest MAE was observed for the Waldstein (summer) campaign (average of 0.17±0.03 m$^2$ g$^{-1}$), indicating that freshly emitted biogenic aerosols are only weakly absorbing. In contrast, a strong relationship was found between the light absorption properties and the concentrations of levoglucosan, corroborating findings from other studies. Regarding the particulate light absorption at 370 nm, a mean particulate light absorption coefficient $b_{abs,370}$ of 54 Mm$^{-1}$ and 6.0

Mm$^{-1}$ was determined for the TROPOS (winter) and Waldstein (summer) campaigns, respectively, with average contributions of particulate brown carbon to $b_{abs,370}$ of 46% at TROPOS (winter) and 15% at Waldstein (summer). The absorption Ångström exponent of the ambient aerosol during the campaigns at TROPOS (winter) and Waldstein (summer) was found to be 1.5±0.1





and 1.2±0.3, respectively. Thus, the Aethalometer measurements support the findings from aqueous filter extracts of only weakly absorbing biogenic aerosols in comparison to the more polluted and BB influenced aerosol at TROPOS (winter).

The mean contribution of NACs to the aqueous extract light absorption over all campaigns ranged from 0.10 %–1.25 % under acidic conditions and 0.13 %–3.71 % under alkaline conditions. The high variability among the measurement sites showed

that the emission strengths of light absorbing compounds and the composition of brown carbon were very different for each site. The mean contribution of NACs to the particulate brown carbon light absorption was 0.10±0.06 % (acidic conditions) and 0.13±0.09 % (alkaline conditions) during the Waldstein (summer) campaign and 0.25±0.21 % (acidic conditions) and 1.13±1.03 % (alkaline conditions) during the TROPOS (winter) campaign. A correlation of NAC concentrations with $Abs_{370}$ was observed for the BB-influenced campaigns at TROPOS (winter) and Melpitz (winter).

The average contribution of NACs to the aqueous extract light absorption over all campaigns was found to be 5 times higher than their mass contribution to water-soluble organic carbon indicating that even small amounts of light-absorbing compounds can have a disproportionately high impact on the light absorption properties of particles.

## 1. Introduction

Organic components of atmospheric aerosols usually treated as solely light scattering  over the near-ultraviolet and visible

range (UV/Vis) of the solar spectrum and therefore only make a negative contribution to radiative forcing (Feng et al., 2013). However, the existence of light absorbing organic carbon (OC) has become more and more evident in the past decade and can be a regionally important phenomenon (see reviews of Andreae and Gelencser (2006) and, more recently, Laskin et al. (2015) and references therein and Ulevicius et al. (2010)). In contrast to black carbon (BC), which absorbs light efficiently over the whole visible and UV region, light absorbtion by OC exhibits a distinct wavelength dependence. The light absorption sharply

increases with decreasing wavelength, making it an efficient absorber in the UV/Vis range. Due to its characteristic yellowish to brownish color, light absorbing OC is also often addressed as brown carbon (BrC).

The light absorption by BrC over the whole solar spectrum are found to be relatively weak compared to BC (Liu et al., 2013a). Nevertheless, at near UV/Vis wavelengths (300-500 nm) BrC has a non–negligible effect on radiative forcing and the regional and global climate (e.g., Bahadur et al., 2012; Feng et al., 2013; Jo et al., 2016; Park et al., 2010). For instance, modelling

studies showed that the radiative forcing of BrC relative to BC is up to 25 % (Feng et al., 2013). Furthermore, BrC light absorption in the UV may alter the concentrations of atmospheric oxidants due to reduced photolysis rates (Jacobson, 1999). The light absorption of ambient particles is generally quantified by the determination of the particulate light absorption coefficient $b_{abs}$, (in Mm$^{-1}$), which can be normalized by the sample mass to give the mass absorption efficiency (MAE in m$^2$ g$^{-1}$). Mass absorption efficiencies can then further be used to estimate the radiative forcing of particulate BrC, which makes it

an important parameter for modelling studies. These models start with modeled emitted mass, which is converted into concentrations using a dispersion model. The absorption of aerosols in the atmosphere is then determined using the appropriate MAE (Feng et al., 2013). The wavelength dependence of light absorption of a particle sample is described by the absorption Ångström exponent (AAE) based on the power-law dependence of $b_{abs}$:

$$\frac{b_{abs,\lambda_1}}{b_{abs,\lambda_2}} = \left(\frac{\lambda_1}{\lambda_2}\right)^{-AAE},$$ (1)

Black carbon has a weak wavelength dependence and often an AAE of 1.0 is assumed, which is  the Mie theoretical value, though measured values range between 0.8–1.1 (Gyawali et al., 2009). Values greater than 1.1 indicate a stronger wavelength dependence, and give evidence for the presence of absorbers at lower wavelengths, such as BrC but also mineral dust (Wang et al., 2013).

Several methods exist to quantify the light absorption properties of BrC. Filter-based and multi-wavelength measurement

techniques are commonly used to assess the contribution of BrC to the total aerosol light absorption (Moosmüller et al., 2009). One of these instruments is the seven-wavelength Aethalometer, which operates from 370 nm to 940 nm. Due to the main light



absorption of BrC at short wavelengths, the 370 nm channel is of particular interest. To estimate the contributions of BrC and BC to aerosol light absorption, a simple approach based on wavelength pairs and assumed AAEs to distinguish between the light absorption of BC and BrC has often been used (Lack and Langridge, 2013).

Another method is to extract aerosol particles into solvents, mostly water or methanol (e.g., Bosch et al., 2014; Chen and Bond, 5 2010; Cheng et al., 2011; Liu et al., 2013a). The advantage of this approach is that BC is insoluble in these solvents and therefore removed by filtration. The extracts are then further analyzed by UV/Vis spectrophotometry. Commonly, light absorption at 365 nm is used to characterize water-soluble BrC. This wavelength is in a range where the light absorption of inorganics does not interfere, whereas the light absorption of organics shows a sufficiently high intensity (Hecobian et al., 2010). Observed MAE values (related to the water-soluble OC, WSOC) for water-soluble BrC typically range between 0.41 10 and 1.80 $m^2 g^{-1}$ (Cheng et al., 2016; Hecobian et al., 2010), where higher values are usually found in winter.

Up till now, field measurements of the light absorption properties of water-soluble BrC have been performed mostly in the US and Asia, and measurements in Europe are scarce. Duarte et al. (2005) investigated the optical properties of WSOC from aerosol samples collected at a rural site in Portugal. They found a seasonal variation with a higher absorptivity in autumn than in summer. Some studies have reported the optical properties of humic-like substances (HULIS), which are a part of WSOC 15 (e.g., Baduel et al., 2010; Utry et al., 2013).

Sources of BrC are very diverse. Known primary sources are biomass burning (BB, Kirchstetter and Thatcher, 2012; Lack et al., 2013; Mohr et al., 2013), and fossil fuel and residential coal combustion (Bond, 2001; Olson et al., 2015). Moreover, BrC can be produced by secondary reactions of anthropogenic (Lin et al., 2015a; Nakayama et al., 2010) or biogenic precursors (Flores et al., 2014; Kampf et al., 2012; Lin et al., 2014).

20 The molecular composition of BrC still remains largely unknown due to its complex nature. Several attempts have been made to characterize BrC at a molecular level (Lin et al., 2015a, 2015b). Among the identified compound groups are large macromolecules, like HULIS (Hoffer et al., 2006), aldol condensation products (Noziere and Esteve, 2005, 2007), and nitrogen-containing compounds formed by the reaction of atmospheric aldehydes with ammonia or amines, e.g. imidazoles (Galloway et al., 2009; Lin et al., 2015b; Yu et al., 2011). Nitrated aromatic compounds (NACs), which are the focus of the 25 present study, comprise another group of major contributors to BrC (Jacobson, 1999; Mohr et al., 2013). Nitrophenols (NP) can be either emitted directly into the atmosphere, e.g. by traffic exhaust (Nojima et al., 1983; Tremp et al., 1993) and wood burning (Hoffmann et al., 2007) or secondarily formed by the nitration of precursor compounds like phenol either in the gas phase or liquid phase (Bolzacchini et al., 2001; Vione et al., 2002). Nitrophenols have been quantified at many different locations, especially in Europe (Cecinato et al., 2005; Iinuma et al., 2010; Kahnt et al., 2013; Zhang et al., 2010), but there is 30 surprisingly little information on NP concentrations in Asia (Chow et al., 2015). Nitrated salicylic acids (NSAs) have also been recently detected in atmospheric aerosol particles (Kitanovski et al., 2012; van Pinxteren and Herrmann, 2007). A strong correlation with nitrate was found by Kitanovski et al. (2012) in samples from Ljubljana, Slovenia, suggesting secondary formation from precursor compounds such as salicylic acid, which has been found in biomass burning aerosols as a lignin degradation product (Iinuma et al., 2007).

35 Studying BrC at a molecular level is considered important, since even trace levels of a compound could have a significant impact on the light absorption properties of particles (Kampf et al., 2012). However, little is known about the contributions of specific light absorbing compounds to the light absorption of either ambient aerosols or aqueous extracts from ambient particles. Mohr et al. (2013) estimated the relative contribution of NACs to particulate BrC light absorption to be about 4 % at 370 nm at a measurement site in the United Kingdom with high influence of BB aerosols. Zhang et al. (2013) calculated a contribution 40 of NACs to aqueous extract light absorption of 4 % from the Los Angeles basin (USA).

The present study aims to expand the understanding of BrC and water-soluble BrC by investigating its spatial and temporal variation in different, very diverse environments (urban, rural, biogenic, high BB influence) including the contributions of individual light-absorbing organic compounds.





## 2. Experimental

### 2.1 Measurement campaigns

Measurements took place at five sampling sites (three German and two Chinese), in six different atmospheric conditions. An overview over the sites, sampling periods, geographical coordinates, and the type of atmospheric environment is given in Table
1. Average temperatures and wind speeds are given as campaign averages.

The Leibniz Institute for Tropospheric Research (TROPOS (winter) campaign, 2014) is located in Leipzig in eastern Germany and can be regarded as a moderately polluted urban background site impacted by a mixture of distributed sources (van Pinxteren et al., 2016). During the measurement period this site was strongly influenced by BB aerosols, as indicated by high levoglucosan concentrations. The first half of the campaign (until 31 January) was characterized by low 12 h mean
temperatures ≤0 °C. Afterwards, the temperature gradually increased towards the end of the campaign (up to 10 °C as 12 h mean). The periods of lower temperatures often coincided with increased emissions from residential heating. The average wind speed was 2.2 m s$^{-1}$.

The measurement site Waldstein in the Bavarian Fichtel Mountains (Waldstein (summer) campaign, 2014, Plewka et al. (2006)) is located about 180 km south west of Leipzig in a low mountain range. The site is surrounded by forest, where spruce is the
dominant species. Measurements took place on a tower at a height of 21 m (about 780 m above sea level). The measurement period was characterized by low wind speeds (<1.3 m s$^{-1}$) and predominantly sunny weather (average temperature of 19 °C). Because of the surrounding forest, the influence of freshly emitted biogenic organics is expected to be high.

Measurements at Melpitz, a rural background site, located about 50 km north east of Leipzig, were carried out in winter (Melpitz (winter) campaign, 2014) and summer (Melpitz (summer) campaign, 2014). The Melpitz (winter) campaign was
characterized by an average temperature of -3 °C and an average wind speed of 2.8 m s$^{-1}$. The Melpitz (summer) campaign was characterized by an average temperature of 22 °C and a mean wind speed of 2.0 m s$^{-1}$. The dates in winter and summer correspond to the measurement periods of the TROPOS (winter) and Waldstein (summer) campaigns, respectively. Due to the proximity of TROPOS and Melpitz, the two sites are influenced by similar regional air masses.

The Chinese measurement sites Xianghe (Xianghe (summer) campaign, 2013) and Wangdu (Wangdu (summer) campaign,
2014) are both located in the Hebei Province in the North China Plain (NCP). The campaigns were part of the CAREBeijing-NCP campaigns in 2013 and 2014. Xianghe is situated between the two megacities of Beijing and Tianjin, and Wangdu is located 170 km south west of Beijing. The average temperature and wind speed at the Xianghe (summer) campaign were 26 °C and 0.8 m s$^{-1}$, respectively. For the Wangdu (summer) campaign a mean temperature of 26 °C and a mean wind speed of 4 m s$^{-1}$ was measured. Both sites can be regarded as regional background stations for the NCP. In comparison to the summer
campaigns in Germany, the average PM$_{10}$ (particulate matter with an aerodynamic diameter ≤ 10 µm) concentrations were about three to seven times higher at the Chinese sites.

### 2.1 Sampling, chemical analysis and back trajectories

Overall, eight NACs were determined as part of BrC: 3-nitrosalicylic acid (3NSA) and 5-nitrosalicylic acid (5NSA), 4-nitrophenol (4NP), 2-methyl-4-nitrophenol (2M4NP), 3-methyl-4-nitrophenol (3M4NP), 2,6-dimethyl-4-nitrophenol
(2,6DM4NP), 2,4-dinitrophenol (24DNP) and 3,4-dinitrophenol (34DNP). The standard compounds were purchased in high purity (≥98%) from either Fluka or Sigma–Aldrich (Munich, Germany).

PM10 was collected on quartz fiber filters with a Digitel DHA-80 high volume filter sampler (MK 360, Munktell, Falun, Sweden, flow rate: 0.5 m$^3$ min$^{-1}$). Day and night samples (11 h or 12 h, see Table S1) were taken during each campaign except for Melpitz (winter) and Melpitz (summer), where particles were collected for 24 h. Details on the sampling times are given
in Table S1. After sampling, filters were stored at -20 °C until extraction. Analysis of WSOC, levoglucosan, NACs and the determination of UV/Vis spectra were carried out using aqueous filter extracts of different portions of filter in ultrapure water.





Details on the different methods are given below. The aqueous filter extract was always filtered through a pre-cleaned syringe filter (0.45µm, Acrodisc 13, Pall, Dreieich, Germany).

Nitrated aromatic compounds and UV/Vis spectra were determined from the aqueous extract by extracting 11–28 pieces of the filter (1.54 cm² each) into 10 mL of ultrapure water. Nitrophenols were analyzed according to the method described in Teich

et al. (2014) based on hollow fiber liquid–phase micro extraction for analyte enrichment and capillary electrophoresis electrospray ionization mass spectrometry (CE-ESI-MS, Agilent $^{3D}$CE instrument, Bruker Esquire 3000$^{+}$ ion trap mass spectrometer). Samples from the Waldstein (summer) campaign were not analyzed for NPs, due to the limited availability of filter material.

Nitrated salicylic acids were enriched by evaporating an alkalinized aliquot of the aqueous filter extract (1.8 mL plus 200 µL

10 mM NaOH) to dryness in a vacuum concentrator (miVac, Genevac Ltd., UK) and redissolving the residue into 40 µL ultrapure water. Subsequent analysis was carried out by CE-MS as described in van Pinxteren et al. (2012). Standard addition was carried out for quantifying the NSA compounds by co-injecting a standard solution plug at different concentration levels into the CE capillary after the sample plug.

As a typical BB tracer, levoglucosan can help to investigate the influence of BB aerosols on the observed concentrations

(Iinuma et al., 2009; Simoneit, 2002). Levoglucosan was analyzed with a Dionex ICS-3000 system coupled with a pulsed amperometric detector (Thermo Fisher Scientific, Sunnyvale, CA, USA). A portion of sampled filter (9.42 cm²) was extracted in 20 mL of ultrapure water by shaking with a laboratory orbital shaker for 120 min. Detailed chromatographic conditions and the merits of analysis can be found elsewhere (Iinuma et al., 2009). Levoglucosan data is available for the campaigns Xianghe (summer), Wangdu (summer) and TROPOS (winter). No data is available for the Melpitz (winter), Melpitz (summer) and

Waldstein (summer) campaigns, due to limited amount of filter material.

Organic carbon and elemental carbon (EC) were determined from the filter by a thermal-optical method using the Sunset Laboratory Dual-Optical Carbonaceous Analyzer (Sunset Laboratory Inc., Tigard, OR, USA) following the EUSAAR 2 temperature-protocol and applying a charring correction using light transmission (Cavalli et al., 2010).

The aqueous extract for determining WSOC was prepared by extracting 21.5 cm² of the filter samples from the Xianghe

(summer), Wangdu (summer), Melpitz (winter) and Melpitz (summer) campaign and 38.5 cm² for samples from the campaigns Waldstein (summer) and TROPOS (winter) into 25 mL and 30 mL of ultrapure water, respectively, followed by 20 min of ultrasonication. After filtration, the extract was injected into a TOC-V$_{CPH}$ analyzer (Shimadzu, Japan) operating in the NPOC (nonpurgeable organic carbon) mode. More details on the method can be found in van Pinxteren et al. (2009).

The air mass origin was estimated by 96 h back trajectories calculated using the HYSPLIT model (Draxler and Rolph, 2003).

**2.2 Instrumentation for aerosol light absorption measurements**

Particulate light absorption was measured by a seven–wavelength Aethalometer, model AE33 (Aerosol d.o.o., Slovenia) during the TROPOS (winter) and Waldstein (summer) campaigns. A detailed description of the instrument is given in Drinovec et al. (2015). Briefly, aerosol particles are collected on a filter tape. Light attenuation is measured continuously through this aerosol laden filter (time resolution of 1s). When a fixed attenuation threshold is reached, the tape advances to a new filter spot. Filter-

based measurements feature non-linear loading effects, caused by the increasing deposition of the sample in the filter loading during the measurement, and filter matrix light scattering effects (Weingartner et al., 2003). The Aethalometer AE33 measures the loading effect by using a dual–spot approach, where attenuation measurements are carried out simultaneously on two differently loaded spots. Multiple scattering effects were compensated by normalizing the particulate light absorption coefficient $b_{abs}$ against data from a Multi–Angle Absorption Photometer (MAAP, Petzold and Schonlinner (2004)) which was

also located at the measurement sites. The determined normalization factor (for light scattering effects) $C$ was 1.69 for the TROPOS (winter) campaign and 2.06 for the Waldstein (summer) campaign. More details are given in the supplement Sect. S1.





The relative contribution of BrC and BC to $b_{abs,370}$ was determined following the method used by Kirchstetter and Thatcher (2012). For the calculations, it was assumed that BC was the only absorbing species at 940 nm, OC and BC were externally mixed, and that the AAE of BC was 1.0. These assumptions are consistent with recently published studies (Martinsson et al., 2015; Mohr et al., 2013). Using Eq. (1), $b_{abs,370,BC}$ was determined by extrapolating from 940 nm to 370 nm, and the contribution of BrC was then calculated using Eq (2):

$$b_{abs,\lambda} = b_{abs,\lambda,BrC} + b_{abs,\lambda,BC} \qquad (2)$$

The AAE was fitted over all seven wavelength and is given as the average of the wavelength dependencies with the measurement time resolution.

### 2.3 UV/Vis spectrophotometry

Light absorption of aqueous solutions were measured with a Lambda 900 UV/Vis-spectrometer (Perkin Elmer) using quartz cells (Secomam, France). The aqueous filter extracts were analyzed at pH≈2 (acidified with $H_2SO_4$, in the following also indicated with the subscript "A") and at pH≈10 (addition of NaOH, in the following also indicated with the subscript "B"). Hence, the target compounds were either in their neutral or deprotonated state. The influence of the pH on MAE and the aqueous light absorption coefficient (Abs) is discussed in Sect. 3.1.1. Spectra were recorded from 300 to 800 nm.

The interpretation of the aqueous extract light absorption follows the method described by Hecobian et al. (2010). In general, according to Beer–Lambert–Law the absorbance ($A_\lambda$) of a solution is defined as:

$$A_\lambda = -\log_{10}\left(\frac{I}{I_0}\right) = l \cdot c \cdot \varepsilon_\lambda = l\sum_i\left(c_i \cdot \varepsilon_{\lambda,i}\right), \qquad (3)$$

where $I_0$ and $I$ are the intensity of the incident and transmitted light, respectively, $l$ the absorbing path length, $c$ the concentration of absorbing species in solution and $\varepsilon_\lambda$ is the wavelength dependent molar extinction coefficient. The resulting data was then converted to the aqueous light absorption coefficient $Abs_\lambda$.

$$Abs_\lambda[Mm^{-1}] = (A_\lambda - A_{800})\frac{V_l}{V_a \cdot l} \cdot ln(10) , \qquad (4)$$

where $A_{800}$ is the reference wavelength to account for any baseline drift, $V_l$ is the volume of water used for extraction (10 mL), $V_a$ is the volume of air passed through the filter, and ln(10) was used to convert the common logarithm to a natural logarithm. The path length l was either 2 or 5 cm.

$Abs_\lambda$ is the light absorption coefficient of the aqueous extract solutions and is not to be mistaken for $b_{abs}$, the light absorption coefficient of ambient particles. This nomenclature was chosen in accordance with other studies (Liu et al., 2013a, 2015).

Using the mass concentration of WSOC (in µg m⁻³), MAE was calculated by:

$$MAE[m^2 g^{-1}] = \frac{Abs_\lambda}{[WSOC]} \qquad (5)$$

The molar extinction coefficient at 370 nm, $\varepsilon_{370}$, was determined for each target compound by preparing a dilution series of standard compounds in water. Subsequently, a three point curve was recorded twice. According to the Beer–Lambert–Law, the slope of the regression in an absorbance–concentration–plot is proportional to $\varepsilon$. The molar extinction coefficients for each compound under acidic and alkaline conditions are presented in Table S3.

In other studies dealing with aqueous extracts, Abs values were given for 365 nm (e.g., Bosch et al., 2014; Cheng et al., 2011; Hecobian et al., 2010). We chose 370 nm to characterize the aqueous extract light absorption properties to match the 370 nm channel of the Aethalometer.

### 2.4 Calculation of the contribution of nitrated aromatic compounds to light absorption of aqueous extracts and particulate brown carbon



The contribution of the target compounds to the aqueous extract light absorption was computed by first calculating the absorbance of the single compound using Eq. (3) and then determining the percentage on the total aqueous solution light absorption.

To assess the contribution of the NACs to particulate BrC light absorption at 370 nm, the molar extinction coefficient of each
compound was converted to the liquid phase molecular absorption cross section $b_{liq}$ (Jacobson, 1999):

$$b_{liq}[cm^2 molecule^{-1}] = 1000 \ln(10)\frac{\varepsilon_{370}}{N_A} \qquad (6)$$

where $N_A$ is the Avogadro constant and $\ln(10)$ is used to convert the common logarithm of the molar extinction coefficient (base 10) to the natural logarithm. The $MAE_{comp}$ of each compound was then calculated by dividing $b_{liq}$ by the mass of one molecule. The subscript "comp" denotes the specific compound. If the $MAE_{comp}$ and the mass concentration in air of the
compound are known, $b_{abs,comp}$ of the single compound in ambient particles can be calculated by the following equation:

$$b_{abs,comp} = MAE_{com} \cdot [comp] \qquad (7)$$

where [comp] is the mass concentration of the compound in the aerosol.

By knowing $b_{abs,comp}$ and $b_{abs,BrC}$, the relative contribution of the individual compound on the particulate BrC light absorption can be calculated.

**3. Results/Discussion**

**3.1. Optical properties of water-soluble and particulate brown carbon**

In the following section, the general optical properties of water-soluble and particulate BrC are presented. Optical properties of water-soluble BrC (represented by the aqueous extract light absorption) are given as $MAE_{370}$ and the aqueous light absorption coefficient ($Abs_{370}$). Particulate BrC was characterized by the particulate light absorption coefficient ($b_{abs,370}$) and
the AAE. The results of the optical properties are summarized in Table 2.

**3.1.1 Influence of acidic and alkaline conditions on the mass absorption efficiency and the aqueous light absorption coefficient**

Values for MAE and Abs have usually been directly derived from filtered aqueous filter extract without considering the pH of the solutions (e.g., Bosch et al., 2014; Cheng et al., 2011; Hecobian et al., 2010). However, depending on the nature of the
absorbing species, pH alters the absorptivity. Alif et al. (1987,1990,1991) showed for NPs that their anionic form is a much stronger absorber than their neutral form. Moreover, the absorption maximum shifts towards longer wavelengths under alkaline conditions. The dissociation in a solution depends on the $pK_a$ value of the compound and at different pH levels different mixtures of neutral/ionized compounds exist. Generally, ambient particles are estimated to be more acidic (pH<4, Scheinhardt et al., 2013), making the findings under acidic conditions more relevant for ambient conditions. However, Hinrichs et al. (2016)
reported, that NPs adsorbed to a particle surface exhibit a significant red-shift similar to the aqueous light absorption spectrum of their deprotonated form. Hence, under certain conditions the deprotonated forms can also be important.

Comparing alkaline and acidic conditions, strong correlations have been found for Abs, MAE and the relative contribution of the individual compounds to Abs over all campaigns ($R^2 \geq 0.93$, see Fig. S3). On average, $Abs_{370}$ increased by a factor of 1.4 and MAE by a factor of 1.6 under alkaline conditions compared to the acidic conditions. As indicated by the good correlation,
this factor is quite consistent and independent of the measurement site.

The molar extinction coefficient at 370 nm increases with pH for the individual compounds (see Table S3). A higher pH leads to deprotonation of hydroxyl and carboxyl groups, which may lead to a shift of the absorption maximum towards longer wavelengths. The light absorption of the aqueous extracts is a sum of many different light absorbing species. Therefore, the relative contribution to the aqueous extract light absorption for individual compounds is affected differently. For example, the
relative contribution of 3NSA and 2,6DM4NP to $Abs_{370}$ decreases at pH 10. No difference in the contribution to $Abs_{370}$ was





observed for 5NSA. The highest increase in the contribution to Abs$_{370}$ was found for 2,4DNP with a factor of 6.8. The observed differences in the contribution to Abs$_{370}$ are related to both the molar extinction coefficient, which alters with the pH, and the composition of the solution. A decrease in the relative contribution to the light absorption at higher pH is most likely due to greater contributions of other, now overlapping light absorbing compounds at this specific wavelength.

The increase of Abs$_{370}$ with higher pH is relatively modest (a factor of 1.4) compared to the increase of the molar extinction coefficient of individual NACs (an average factor of 4.5). This might indicate that the absorption properties of the majority of light absorbing compounds are little affected by the pH. These findings are only valid for the investigated wavelength of 370 nm and are likely to differ over the whole spectral range.

Due to the close relationship between MAE and Abs under alkaline and acidic conditions, only the values for the acidic

conditions are discussed in the following section (Sect. 3.1.2), while both of them are always displayed in the Figures and Tables.

**3.1.2. Mass absorption efficiency and aqueous light absorption coefficient of water-soluble brown carbon**

The temporal evolution of Abs$_{370}$ and MAE$_{370}$ for each measurement site are shown in Fig. 1. The highest mean MAE$_{370A}$ values were found for the campaigns Melpitz (winter, $0.86\pm0.33$ m$^2$ g$^{-1}$) and TROPOS (winter, $0.84\pm0.20$ m$^2$ g$^{-1}$) followed by Wangdu

(summer, $0.55\pm0.15$ m$^2$ g$^{-1}$) and Xianghe (summer, $0.38\pm0.06$ m$^2$ g$^{-1}$). The lowest values were observed under summer conditions in Germany, where the values for the Melpitz (summer) campaign slightly exceeded the MAE$_{370A}$ observed at the Waldstein (summer) campaign ($0.22\pm0.03$ m$^2$ g$^{-1}$ over $0.17\pm0.03$ m$^2$ g$^{-1}$). Measured MAE$_{370A}$ values for China are in the same range or slightly lower than values reported in other studies. For instance, several studies were conducted in Beijing in summer, where Cheng et al. (2011) determined an average MAE of 0.7 m$^2$ g$^{-1}$, Du et al. (2014) found 0.5 m$^2$ g$^{-1}$ and Yan et al. (2015)

observed 0.7 m$^2$ g$^{-1}$ (all values determined at 365 nm). The findings for the TROPOS (winter) and Melpitz (winter) campaigns suggest regional behavior and are comparable to observed MAEs from urban sites in the US (Hecobian et al. (2010), 0.47-0.87 m$^2$ g$^{-1}$, annual mean, determined at 365 nm). Low MAE values, similar to the observed MAEs at the Melpitz (summer) and Waldstein (summer) campaigns, have been found at urban and rural sites in the U.S. (0.21 and 0.12 m$^2$ g$^{-1}$, respectively, determined at 365 nm) when the concentration of levoglucosan was low (<50 ng m$^{-3}$). Although the average Abs$_{370A}$ is similar

for the campaigns of Wangdu (summer), TROPOS (winter) and Melpitz (winter) (4.64, 3.92 and 4.13 Mm$^{-1}$, respectively), the observed mean MAE$_{370A}$ for the Wangdu (summer) campaign is lower, indicating that despite higher concentrations of OC and WSOC during the Wangdu (summer) campaign (see Table S4), fewer light absorbing compounds were present.

A correlation between levoglucosan and Abs$_{370}$ was found during the TROPOS (winter) campaign (R$^2$ = 0.8), and during the BB episode of the Wangdu (summer) campaign (R$^2$ = 0.9), when BB aerosol was abundant (see Fig. 2). This correlation

suggests that BB aerosol is a major contributor to the aqueous extract light absorption. The connection between BB and light absorbing BrC was also observed in a number of other studies (e.g., Desyaterik et al., 2013; Hoffer et al., 2006; Lack et al., 2013; Mohr et al., 2013). During non-BB episodes of the Wangdu (summer) campaign and the Xianghe (summer) campaign, only a weak or no correlation at all was observed between levoglucosan and Abs$_{370}$ (see Fig. 2). This indicates the presence of sources for light absorbing compounds other than BB aerosols. A decay of levoglucosan more rapid than the decay of NACs

might also be a cause for this observation (Hoffmann et al., 2010).

Regarding seasonal differences, the aqueous extract light absorption was much lower in summer compared to winter samples. This is a general trend, observed at many measurement sites (e.g., Du et al., 2014; Hecobian et al., 2010; Kim et al., 2016). Kim et al. (2016) suggested that this is either because the amount of emitted BrC is much higher during winter or that summer BrC is less water-soluble. By comparing aqueous extract light absorption to ambient particle light absorption measurements

this issue will be further explored and discussed in the following section.

**3.1.3 Particulate light absorption coefficient and absorption Ångström exponent of particulate brown carbon**



The temporal evolution of the measured particulate light absorption coefficient $b_{abs}$ at 370 nm as well as the determined contributions of particulate BrC and BC to the light absorption at 370 nm are shown in Fig. 3. The AAE (as campaign average) obtained by fitting over all Aethalometer wavelengths was 1.5±0.1 and 1.2±0.3 for the campaigns TROPOS (winter) and Waldstein (summer), respectively. An AAE close to one indicates that BC is the dominant species (Kirchstetter et al., 2004).

5 The increased AAE values indicate that BrC plays a larger role for the TROPOS (winter) campaign than for the Waldstein (summer) campaign, where its contribution is minor.

During the BB influenced TROPOS (winter) campaign, the mean total $b_{abs,370}$ was 54 Mm$^{-1}$. Particulate BrC contributed about 40 % to the overall light absorption at 370 nm. The influence of BB on the light absorption of ambient particles has also been observed in other studies. For instance, Kirchstetter and Thatcher (2012) estimated that the contribution of BrC to the light

10 absorption of wood smoke is 49 % below 400 nm from samples collected in wintertime California (U.S.). Mohr et al. (2013) found a BrC contribution of 46% to $b_{abs,370}$ at a UK site that was highly influenced by BB. Our findings on the contribution of BrC to $b_{abs,\ 370}$ during the TROPOS (winter) campaign are consistent with these earlier measurements. The light absorption was observed to be higher for easterly air masses than those from the west, a trend also seen in the NAC concentrations of the TROPOS (winter) campaign (see Sect. 3.2). Easterly winds were often accompanied by lower temperatures or thermal

15 inversion layers or both. Therefore, higher concentrations of light absorbing compounds (and higher particulate light absorption) might be a result of both, increased emissions (locally and in the source region) and lower dispersion of absorbing pollutants due to the thermal inversion layer.

In contrast, the light absorption of aerosol particles was much lower during the Waldstein (summer) campaign with a mean total $b_{abs,370}$ of 6 Mm$^{-1}$ and a BrC contribution to the particulate light absorption of about 15 % at 370 nm. The measurements

20 at Waldstein (which is surrounded by a mixed forest) were carried out in a sunny summer period when high biogenic emissions would strongly influence the results. Several laboratory studies have shown that the formation of light absorbing compounds from biogenic precursors is negligible (e.g., Liu et al., 2013b; Nakayama et al., 2010; Song et al., 2013). However, a few studies have suggested that BrC might be formed in the presence of high ammonia concentrations (Flores et al., 2014; Updyke et al., 2012) or from isoprene epoxydiols (IEPOX, Lin et al., 2014). Findings from field measurements in the Amazon Basin

25 confirmed that the light absorption by biogenic aerosol is much lower than the light absorption of BB aerosols or BC (Rizzo et al., 2011). Compared to the TROPOS (winter) campaign, $b_{abs}$ for BrC is very small for the Waldstein (summer) campaign, with an average of 0.94 Mm$^{-1}$ compared to 21.8 Mm$^{-1}$. Our findings corroborate those of Rizzo et al. (2011) and laboratory studies stating that biogenic aerosols alone produce only small amounts of light absorbing compounds.

The relative contribution of light-absorbing WSOC to particulate light absorption can be estimated by applying a conversion

30 factor. Based on Mie theory calculations, Liu et al. (2013a) suggested a factor of two to convert the aqueous extract light absorption coefficient (Abs) into the particulate light absorption coefficient ($b_{abs}$). For our data, this gives a good agreement between the Waldstein (summer) average $Abs_{370A}$ (0.51 Mm$^{-1}$ x 2) and the Waldstein (summer) average $b_{abs,370,BrC}$ (0.94 Mm$^{-1}$), suggesting that the observed particulate light absorption is mainly associated with water-soluble compounds during this campaign. For the TROPOS (winter) campaign, doubling the average $Abs_{370A}$ gives a value of 7.8 Mm$^{-1}$, which is still much

35 lower than the average $b_{abs,370,BrC}$ (21.8 Mm$^{-1}$). Thus, a large fraction of non-water-soluble OC likely exists as well, that is highly light absorbing at 370 nm.

## 3.2. Concentrations of nitrated aromatic compounds under different atmospheric conditions

This section presents an overview and a comparison of the NAC concentrations for all campaigns. Table 3 summarizes the measured concentrations in comparison with other studies at similar locations. The results of the temporal variation of the

40 target compounds for each campaign are displayed in Fig. 1.

A general concentration trend of 4NP > 3M4NP > 2M4NP > 2,6DM4NP > 2,4DNP > 3,4DNP and 3NSA > 5NSA was observed for all campaigns of the present study and is in good agreement with findings from most other studies. Regarding the



sum of NPs, the mean concentrations were highest during the TROPOS (winter) campaign (14.0 ng m$^{-3}$) followed by the Melpitz (winter) campaign (11.1 ng m$^{-3}$). These concentrations are a factor of 3 to 10 higher than those observed during the campaigns Wangdu (summer) and Xianghe (summer) (4.4 ng m$^{-3}$ and 1.4 ng m$^{-3}$, respectively). During the Melpitz (summer) campaign, the NP concentration was 100 times lower (average of 0.1 ng m$^{-3}$) than during the Melpitz (winter) campaign. For

the sum of NSAs, the highest mean concentrations were found at the Wangdu (summer) campaign (4.8 ng m$^{-3}$) followed by the campaigns Xianghe (summer) and TROPOS (winter) (average of 2.2 ng m$^{-3}$, each). Average concentrations for NSAs of $\leq$ 1 ng m$^{-3}$ were observed at the campaigns Melpitz (winter, 1.0 ng m$^{-3}$), Waldstein (summer, 0.4 ng m$^{-3}$) and Melpitz (summer, 0.2 ng m$^{-3}$).

A clear seasonality can be seen in the data from the German sites. The observed concentrations of the different species are

generally lower in summer than in winter because the sources of NACs are stronger during winter and generally connected to polluted air, as discussed in the Sect. 1.

The campaigns Melpitz (winter) and TROPOS (winter) were carried out in the same period and the two sites are directly comparable due to their geographic proximity. Concentrations of the NACs are in the same range at both sites and show a common pattern. Two periods with concentration maxima were found. The first period occurred around 25 January and the

second peak was around 30 January, 2014 (see Fig. 1, a and b). Both periods are characterized by prevailing easterly winds and shorter mean lengths of 96-hour back trajectories. In contrast, air masses during other periods originated from the south west and had longer trajectories and increased residence times over the Atlantic Ocean. Higher concentrations of PM and many of its constituents with easterly winds are often observed at Melpitz (Spindler et al., 2010). The long residence time above the continent results in increased concentrations of pollutants in those air masses. Furthermore, as described in Sect. 3.1.3, the

temperature was generally lower when the air mass originated from the east, which likely led to higher emissions from residential heating both locally and in the air mass source regions.

The campaigns Melpitz (summer) and Waldstein (summer) were also conducted in the same time period. The concentrations of NSAs are comparable at both sites (0.01-0.51 ng m$^{-3}$) and are very low compared to the winter measurements. Nitrophenols were not analysed from the Waldstein (summer) campaign as explained above.

Day and night samples were collected during the campaigns TROPOS (winter), Waldstein (summer), Xianghe (summer) and Wangdu (summer). Nighttime concentrations were found to be slightly higher than during the day. The lower daytime concentrations are probably caused by the higher boundary layer heights during the day, i.e. dilution effects, and/or lower emission/formations rates during daytime.

The sum of all target compounds at the TROPOS (winter) campaign was only weakly correlated with levoglucosan ($R^2 = 0.47$,

Fig. S4), indicating that BB was not the dominant source of NACs during the campaign. As mentioned above, the formation or decay processes of levoglucosan and NACs might be very different, which resulted in no correlation of those species. It is also known, that the sources of NACs are very diverse (see Sect. 1).

At the Wangdu (summer) campaign, a BB episode was observed between 11 June and 19 June (see Fig. 1f), indicated by a peak in measured levoglucosan concentrations. During that period, increased concentrations of NACs were found. Although

the NAC concentration generally increased with increasing levoglucosan concentration, no clear connection between NACs and BB aerosols was found. For the Xianghe (summer) campaign, levoglucosan concentrations (average of 109 ng m$^{-3}$) were lower than for the Wangdu (summer) or TROPOS (winter) campaign, suggesting that BB played a minor role at this site during the time of measurements. No correlation was found between the sum of NACs and levoglucosan. This implies that BB was neither the major source of NACs in the Wangdu (summer) nor in the Xianghe (summer) campaign.

Nitrated aromatic compounds seem to be ubiquitous and are found at each site in measurable amounts although they are highly variable at the different measurement sites. The sum of NAC concentrations at the different campaigns followed the order TROPOS (winter) > Melpitz (winter) > Wangdu (summer) > Xianghe (summer) > Waldstein (summer) > Melpitz (summer). This trend might be considered surprising, since the average PM$_{10}$ and OC values were higher at the Chinese sites than at the





German sites (see Table S4). However, the seasonality and influence of BB aerosol seems to play a large role for the NAC concentration resulting in lower concentrations at the Chinese summer sites than at the German winter sites.

### 3.3. Contribution of nitrated aromatic compounds to water-soluble and particulate brown carbon

In this section, the contribution of NACs to the aqueous extract light absorption and then the contribution of NACs to the particulate BrC light absorption are discussed. The relative contribution of NACs to $Abs_{370}$ is shown in Fig. 4 together with the contribution of NACs to the WSOC concentration. Table 2 summarizes the contribution of NSAs and NPs to aqueous extract light absorption (Abs) and particulate BrC light absorption ($b_{abs}$).

The relative NAC contribution to aqueous extract light absorption varied greatly across the different sites, ranging from 0.02 to 4.41 % for acidic conditions and from 0.02 to 9.86 % for alkaline conditions. This result likely reflects the dependence of NAC source strength on location and seasonality. The maximum contribution of NACs to $Abs_{370}$ was found for the TROPOS (winter) campaign. The contribution of NACs to $Abs_{370}$ was low for the campaigns Waldstein (summer) and Melpitz (summer). For the Waldstein (summer) campaign, a mean contribution of 0.15% (acidic) or 0.13% (alkaline) was observed. Average values of 0.13% (acidic) and 0.24% (alkaline) where obtained from the Melpitz (summer) campaign. The results for the Xianghe (summer) and Wangdu (summer) campaigns are about half the values of the TROPOS (winter) or Melpitz (winter) campaigns. This shows, that NACs in the present study had a much higher impact on the light absorption properties during the German winter campaigns than during the Chinese summer campaigns and indicates that other compounds play a larger role in particle light absorption at the Chinese sites. More studies are needed to resolve the molecular identity and abundance of these light absorbing compounds.

The relative mass contribution of NACs to the WSOC concentration at the different campaigns follows the same pattern as the contribution of NACs to the aqueous extract light absorption and also the same pattern as the NAC concentrations (see Sect. 3.2 and Fig. 4). The highest mass contributions of NACs were found during the TROPOS (winter) campaign followed by the Melpitz (winter) campaign, and lowest values were observed for the campaigns Waldstein (summer) and Melpitz (summer). The average mass contribution of NACs to WSOC is less than 0.2% at each site. In contrast, NACs have a mean contribution to aqueous extract light absorption of 1%, which is a factor of five larger than their mass contribution to WSOC. Our results corroborate the findings of other studies, which show that even small amounts of light absorbing compounds can significantly impact the light absorption properties of particles. As a result, a detailed investigation of the molecular composition of BrC is important (Kampf et al., 2012; Laskin et al., 2015).

The correlation between NAC concentrations and $Abs_{370}$ for each campaign is displayed in Fig. 5. The correlation is good for the TROPOS (winter) and Melpitz (winter) campaign ($R^2$=0.7, each) and weaker for the Wangdu (summer) campaign ($R^2$=0.6). No correlation was found for the campaigns Xianghe (summer), Melpitz (summer) and Waldstein (summer). In general, the presence of high NAC concentrations also indicates higher values for $Abs_{370}$ than at sites with lower NAC concentrations. However, due to the diverse sources and sinks (e.g. photobleaching (Zhao et al., 2015)) of NACs and BrC, NACs cannot be considered as the only tracers for BrC for very different atmospheric conditions.

The average contribution of NACs to particulate BrC light absorption, was found to be 0.10 % for the Waldstein (summer) campaign and 0.25 % for the TROPOS (winter) campaign for the protonated forms, and 0.13 % and 1.18 %, respectively, for their deprotonated forms. The contribution during the Waldstein (summer) campaign is similar for the aqueous extract and aerosol light absorption, which is consistent with the assumption discussed above that the observed light absorption is mostly derived from WSOC and no further water-insoluble BrC was present at this site.

To the best of the author's knowledge, up to now only two other field studies have determined the contribution of individual compounds to BrC light absorption. Zhang et al. (2013) estimated the contribution of eight NACs to aqueous extract light absorption in Los Angeles to be about 4% at 365 nm, while Mohr et al. (2013) estimated a NAC contribution of 4 % to particulate BrC light absorption at 370 nm at a BB influenced site in the U.K. The findings from Zhang et al. (2013) are in the




same range as our findings for the TROPOS (winter) campaign (for $Abs_{370}$), whereas the estimated contribution of NACs to particulate BrC light absorption by Mohr et al. (2013) is a factor of 16 larger than our observed contributions. One possible reason for this discrepancy is that the mean $b_{abs,BrC,}$ measured in the present study is about twice as high as observed by Mohr et al. (2013) (10 Mm$^{-1}$) while the concentrations of NACs are in the same range in both studies. Moreover, Mohr et al. (2013)

considered a different suite of NACs than the present study. Another important point is that the calculation by Mohr et al. (2013) used values based on the absorption maxima of the individual species, which usually lie below 370 nm (for their protonated form). In contrast, in the present study the molar extinction coefficient of each compound was directly determined at 370 nm. When calculations were made based on the absorption maxima of protonated NACs, an average contribution to the particulate BrC light absorption of 1 % was found for the present study, i.e a factor of four higher than the calculation at 370

nm. This suggests that the influence of NACs is higher towards shorter wavelength, i.e. near their absorption maximum and thus can be more important in terms of their influence on atmospheric photochemistry, e.g. $O_3$ photolysis (Jacobson, 1999).

## 4. Conclusions

In the present study, the contributions of eight NACs to the light absorption of aqueous particle extracts and particulate BrC were determined in Germany and China at five measurement sites under six different atmospheric conditions.

The effect of pH on the aqueous extracts light absorption was investigated. Absorption of extracts at 370 nm, $Abs_{370}$, increased by a factor of 1.4 and MAE by a factor of 1.6 when the pH was increased from 2 to 10. A general seasonal trend of $MAE_{370}$ and $Abs_{370}$ being higher in winter than in summer was observed, which is in agreement with other studies. The MAE values during winter times impacted by BB in Germany exceeded those of the Chinese background stations during summer. Very low MAE and $b_{abs,370}$ were observed at the forest site Waldstein in summer, indicating that freshly emitted biogenic aerosols

are only weakly absorbing. In contrast, a strong relationship was found between the light absorption properties and BB aerosol concentrations, corroborating findings from other studies. An average contribution of particulate BrC to $b_{abs,370}$ of 46% was observed during the TROPOS (winter) campaign and 15% during the Waldstein (summer) campaign. The AAE of particulate BrC was found to be 1.5 and 1.2 for the campaigns TROPOS (winter) and Waldstein (summer), respectively.

The relative contribution of NACs to the aqueous extract light absorption was highly variable depending on the measurement

site, ranging from 0.02 % to 4.41 % under acidic conditions and 0.02 % to 9.86 % under alkaline conditions. This indicates that the emission strength of light absorbing compounds and the composition of BrC differed between the sites. In addition, the formation and decay processes might be very different in the respective environments. The mean contribution of NACs to the particulate BrC light absorption was 0.10 % (acidic conditions) and 0.13 % (alkaline conditions) during the Waldstein (summer) campaign and 0.23 % (acidic conditions) and 1.15 % (alkaline conditions) during the TROPOS (winter) campaign.

A correlation between the NAC concentration and $Abs_{370}$ was observed during the campaigns TROPOS (winter) and Melpitz (winter), at other sites the correlation was weak or non-existent.

The mass contribution of NACs to WSOC was five times lower than their contribution to the aqueous extract light absorption. This corroborates conclusions of other studies that even small amounts of light absorbing compounds can have a disproportionately high impact on the aerosol light absorption properties (Kampf et al., 2012).

Field studies on the molecular composition of BrC are still scarce, especially for sites that have no or little influence of BB aerosols. Therefore, more efforts are needed to assess BrC on a molecular level, since a deeper knowledge of the present light absorbing compounds can improve the understanding and prediction of BrC aging processes and its radiative forcing (Laskin et al., 2015). A further investigation of BrC compounds may also result in finding tracer compounds for specific BrC sources.

**Acknowledgements**. This research was part of the "BranKo" project, supported by the German Research Foundation (DFG) under contracts PI 1102/3-1 and HE 3086/26-1. Travelling and accomodation costs for CAREBeijing-NCP 2013 and CAREBeijing-NCP 2014 were funded by the EU project AMIS 295132 and the Sino German Science Center No. GZ663,



respectively. The authors would like to thank Gerald Spindler and Yoshiteru Iinuma, for providing OC/EC and levoglucosan concentrations, respectively. Dean Venables is thanked for language corrections and providing useful suggestions. The analytical work of the technical staff in the laboratories of TROPOS ACD is also acknowledged. G. Močnik is employed by the company Aerosol d.o.o. where the Aethalometer AE33 was developed and is manufactured.

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



**Table 1.** Overview over sampling sites, used terminology, sampling period, sampling duration and according atmospheric conditions during the sampling period.

| Sampling site | Designation | Geographic coordinates | Sampling period (sampling duration per filter) | Comments |
|---|---|---|---|---|
| **Germany** | | | | |
| Roof of the Leibniz Institute for Tropospheric Research, Leipzig | TROPOS (winter) | 51.35 °N, 12.43 °E | 24 January to 08 February 2014 (12 h, day and night) | Urban background, strong influence of biomass burning aerosol during measurement period (van Pinxteren et al., 2016) |
| Melpitz | Melpitz (winter) | | 24 January to 04 February 2014 (24 h) | Rural background (Spindler et al., 2010) |
| | Melpitz (summer) | 51.53 °N, 12.93 °E | 16 July 2014 to 25 July 2014 (24 h) | |
| Bavarian Fichtel Mountains | Waldstein (summer) | 50.14 °N, 11.87 °E | 16 July to 25 July 2014 (11 h, day and night) | BayCEER Waldstein observatory, F–BEACH campaign, surrounded by forest, high local biogenic emissions (Plewka et al., 2006) |
| **China** | | | | |
| Xianghe | Xianghe (summer) | 39.75 °N, 116.96 °E | 09 July to 14 July and 21 July 2013 to 01 August 2013 (12 h, day and night) | CAREBeijing–NCP 2013 campaign (Kecorius et al.., 2015), rural background, urban outflow |
| Wangdu | Wangdu (summer) | 38.71 °N, 115.13 °E | 04 June to 24 June 2014 (12 h, day and night) | CAREBeijing–NCP 2014 campaign, rural background, biomass burning episode occurred during the measurement period (Kecorius et al., 2016) |



**Table 2.** Optical properties and calculated contribution of NACs to the light absorption of aqueous extracts and particulate BrC. The values are given as: minimum-maximum (mean±standard deviation).

| | Waldstein (summer) | Melpitz (summer) | TROPOS (winter) | Melpitz (winter) | Xianghe (summer) | Wangdu (summer) |
|---|---|---|---|---|---|---|
| **Optical properties** | | | | | | |
| $Abs_{370A}$ [Mm$^{-1}$] | 0.21-0.70 (0.51±0.11) | 0.57-1.34 (0.84±0.24) | 0.33-8.96 (3.91±2.49) | 1.75-10.6 (4.13±2.51) | 0.82-4.68 (2.02±0.81) | 1.10-21.8 (4.64±4.16) |
| $Abs_{370B}$ [Mm$^{-1}$] | 0.63-1.15 (0.88±0.13) | 0.88-1.78 (1.19±0.30) | 0.69-14.5 (6.80±3.94) | 3.28-15.2 (6.63±3.50) | 1.20-9.07 (3.16±1.50) | 1.87-27.2 (6.65±5.55) |
| $MAE_{370A}$ [m$^2$ g$^{-1}$] | 0.10-0.25 (0.17±0.03) | 0.19-0.30 (0.22±0.03) | 0.41-1.30 (0.84±0.20) | 0.52-1.79 (0.86±0.33) | 0.30-0.52 (0.38±0.06) | 0.31-1.01 (0.55±0.15) |
| $MAE_{370B}$ [m$^2$ g$^{-1}$] | 0.24-0.36 (0.29±0.03) | 0.27-0.43 (0.31±0.05) | 0.86-2.21 (1.45±0.29) | 0.84-2.57 (1.37±0.45) | 0.34-1.15 (0.59±0.14) | 0.49-1.40 (0.81±0.21) |
| $b_{abs370}$ [Mm$^{-1}$] | 3.72-8.59 (6.23±1.33) | | 4.13-129.4 (54.2±34.07) | | | |
| $b_{abs370,OC}$ [Mm$^{-1}$] | 0.62-1.69 (0.94±0.28) | | 1.25-54.4 (21.8±14.2) | | | |
| **Contribution to light absorption [%]** | | | | | | |
| NSA to $Abs_{370A}$ | 0.02-0.33 (0.15±0.10) | 0.03-0.13 (0.08±0.04) | 7.32·10$^{-4}$-0.19 (0.11±0.06) | 0.01-0.19 (0.06±0.05) | 0.02-1.02 (0.26±0.20) | 0.04-0.89 (0.29±0.19) |
| NSA to $Abs_{370B}$ | 0.02-0.25 (0.13±0.03) | 0.03-0.13 (0.07±0.04) | 5.9·10$^{-4}$-0.15 (0.09±0.05) | 0.004-0.09 (0.05±0.03) | 0.01-0.81 (0.22±0.16) | 0.04-0.71 (0.25±0.15) |
| NSA to $b_{abs370,OC}$ (A) | 0.01-0.22 (0.10±0.06) | | 0.001-0.04 (0.02±0.01) | | | |
| NSA to $b_{abs370,OC}$ (B) | 0.02-0.31 (0.13±0.09) | | 0.001-0.06 (0.03±0.01) | | | |
| NP to $Abs_{370A}$ | | 0.02-0.09 (0.05±0.02) | 0.33-4.22 (1.14±0.82) | 0.14-1.84 (0.81±0.52) | 0.04-0.62 (0.22±0.13) | 0.10-1.19 (0.26±0.17) |
| NP to $Abs_{370B}$ | | 0.07-0.35 (0.17±0.09) | 0.91-9.71 (3.62±2.05) | 0.47-4.90 (2.43±1.42) | 0.17-2.72 (0.83±0.53) | 0.38-3.97 (1.02±0.59) |
| NP to $b_{abs370,OC}$ (A) | | | 0.06-1.13 (0.23±0.21) | | | |
| NP to $b_{abs370,OC}$ (B) | | | 0.28-5.42 (1.15±1.03) | | | |





**Table 3.** Measured concentrations of nitrophenols and nitrated salicylic acids for each campaign in comparison to other studies. The concentrations are given as: minimum-maximum (mean±standard deviation).

| Location | Concentration in ng m$^{-3}$ | | | | | | | | Reference |
|---|---|---|---|---|---|---|---|---|---|
| | 4NP | 2M4NP | 3M4NP | 2,6DM4NP | 2,4DNP | 3,4DNP | 3NSA | 5NSA | |
| **Europe** | | | | | | | | | |
| TROPOS (winter), Germany, Jan-Feb 2014 | 1.08-27.2 (7.09±7.08) | 0.75-12.3 (3.64±3.05) | 0.50-8.58 (2.60±2.22) | 0.11-2.29 (0.65±0.58) | n.d. | n.d. | 0.01-3.72 (1.36±1.02) | 0.02-2.95 (0.94±0.75) | This study |
| Melpitz (winter), Germany, Jan-Feb 2014 | 0.51-10.3 (4.09±3.27) | 0.29-8.90 (3.64±3.06) | 0.15-6.40 (2.44±2.20) | 0.04-2.74 (0.91±0.90) | 0.003-0.01 (0.01±0.00) | n.d. | 0.09-2.30 (0.66±0.69) | 0.06-0.77 (0.32±0.24) | This study |
| Detling, UK, Jan-Feb 2012 | (0.02)[b] | (5.0)[b] | | - | (3.0)[b] | - | (3.0)[b] | | Mohr et al. (2013) |
| Ljubljana, Slovenia, Dec 2010 to Jan 2011 | 0.5-3.7 (1.8) | 0.31-1.5 (0.75) | 0.25-1.2 (0.61) | - | 0.02-0.05 (0.02) | - | 0.1-3.9 (1.3) | 0.2-3.4 (1.4) | Kitanovski et al. (2012) |
| Mainz, Germany, Winter2006/2007 | (5.5)[a] | - | - | - | - | - | - | - | Zhang et al. (2010) |
| Rome, Italy, Feb to Apr 2003 | (17.8) | - | (7.8) | (5.9) | - | - | - | - | Cecinato et al. (2005) |
| Hamme, Belgium, winter 2010/2011 | 0.92-3.0 (1.19)[b] | - | - | - | - | - | - | - | Kahnt et al. (2013) |
| Melpitz (summer), Germany, Jul 2014 | 0.01-0.12 (0.06±0.03) | 0.03-0.04 (0.04±0.00) | 0.02-0.03 (0.03±0.00) | n.d. | n.d. | n.d. | 0.07-0.42 (0.17±0.15) | 0.02-0.24 (0.09±0.09) | This study |
| Waldstein (summer), Germany, Jul 2014 | - | - | - | - | - | - | 0.01-0.35 (0.17±0.11) | 0.06-0.51 (0.23±0.12) | This study |
| Ljubljana, Slovenia, Aug 2010 | 0.12-0.17 (0.15) | <0.03-0.05 (0.05) | <0.03 | - | <0.01 | - | 0.06-0.12 (0.09) | 0.14-0.24 (0.09) | Kitanovski et al. (2012) |
| Mainz, Germany, summer 2006 | (2.8)[a] | - | - | - | - | - | - | - | Zhang et al. (2010) |
| Hamme, Belgium, summer 2010/2011 | 0.17-0.62 (0.29)[b] | - | - | - | - | - | - | - | Kahnt et al. (2013) |
| **China** | | | | | | | | | |
| Xianghe (summer), China, Jul-Aug 2013 | 0.13-4.49 (0.98±0.78) | 0.01-0.85 (0.32±0.21) | 0.004-0.46 (0.09±0.07) | 0.01-0.23 (0.06±0.05) | $2.00 \cdot 10^{-4}$-0.10 (0.02±0.03) | 0.004-0.17 (0.03±0.04) | 0.13-8.99 (1.21±1.45) | 0.11-2.70 (0.88±0.64) | This study |
| Wangdu (summer), China, Jun2014 | 0.55-15.0 (2.63±2.66) | 0.01-4.35 (0.68±0.78) | 0.01-2.03 (0.21±0.35) | $3.54 \cdot 10^{-4}$-0.47 (0.06±0.09) | 0.002-0.41 (0.08±0.10) | 0.003-0.02 (0.01±0.01) | 0.37-11.3 (3.14±3.05) | 0.43-3.27 (1.63±0.78) | This study |
| Hong Kong, summer 2010-2012[c] | (0.3) | (0.2) | (0.02) | (0.009) | - | - | - | - | Chow et al. (2015) |

[a] estimated from graph
[b] sum of corresponding m/z ratios, including all isomers
[c] Concentrations are given as the average of three years

n.d.: not detected





**Figure 1.** Temporal variation of the concentrations of nitrated aromatic compounds and levoglucosan (Levo) at each measurement site (a-f) and their optical properties. The aqueous light absorption coefficient (Abs) and mass absorption efficiency (MAE) are given at 370 nm for acidic conditions (indicated by the subscript "A") and for alkaline conditions (indicated by the subscript "B").





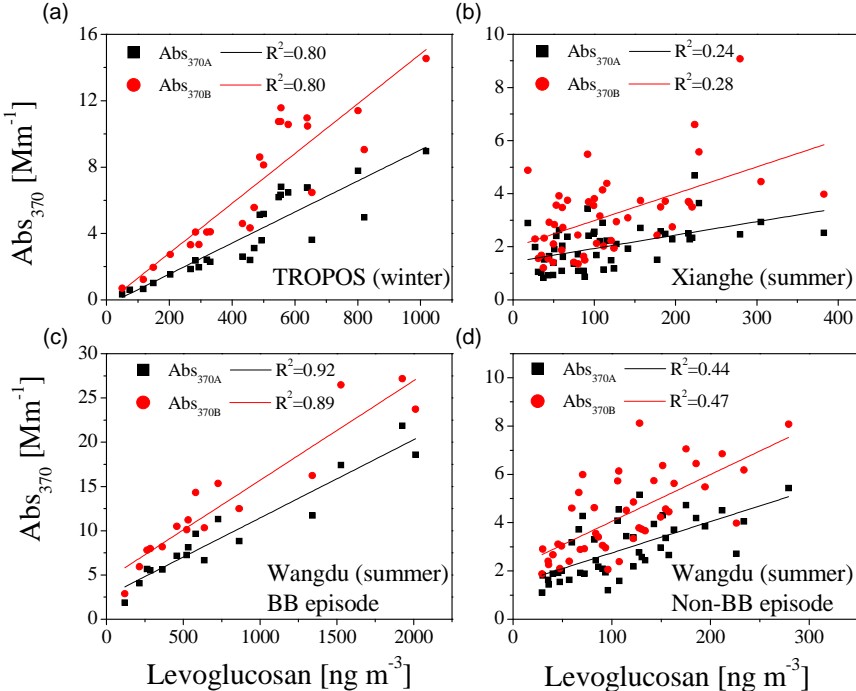

**Figure 2.** Scatter plot of the aqueous light absorption coefficient $Abs_{370}$ with levoglucosan concentration for the campaigns (a) TROPOS (winter,), (b) Xianghe (summer), (c) Wangdu (summer, BB episode) and (d) Wangdu (summer, non-BB episode). $Abs_{370}$ is given for acidic conditions (indicated by the subscript "A", black squares) and for alkaline conditions (indicated by
5  the subscript "B", red dots). Note different axis scales.





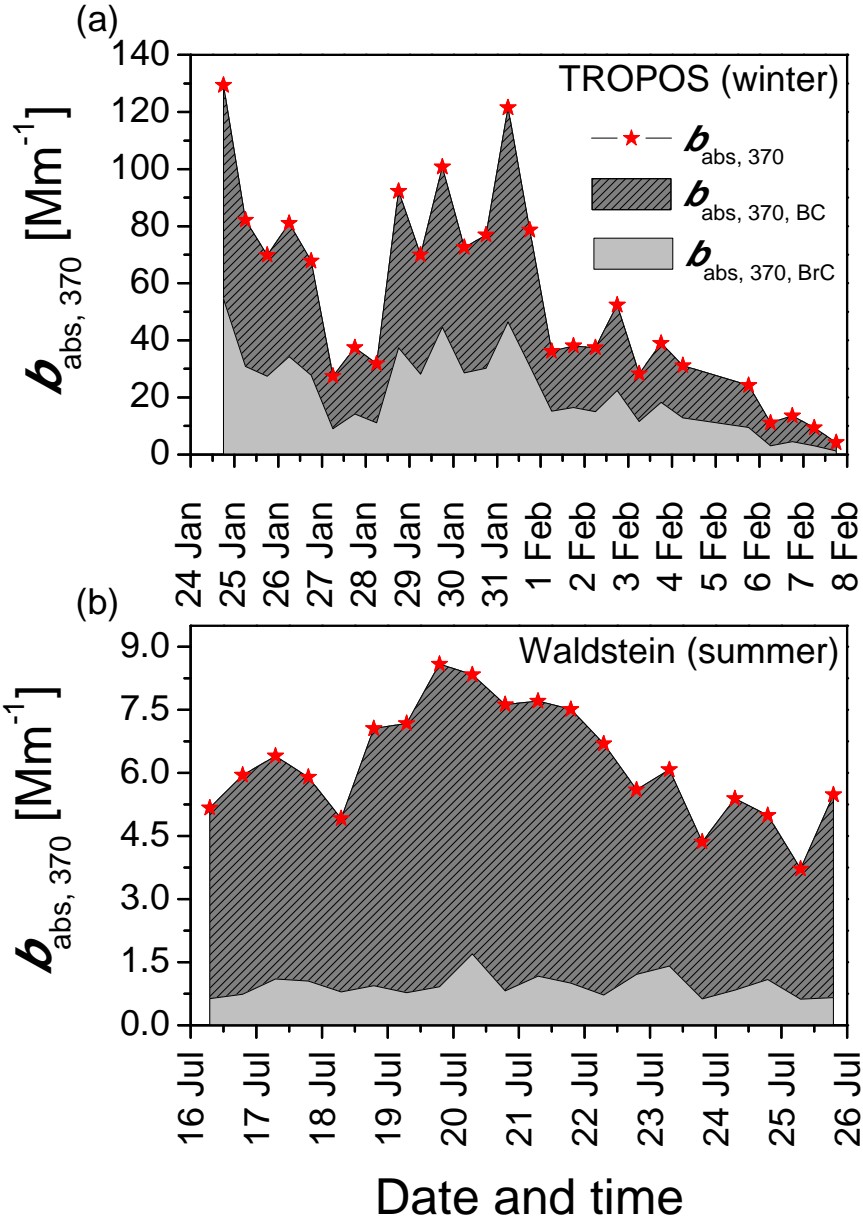

**Figure 3.** Temporal evolution of the particulate light absorption coefficient $b_{abs}$ at 370 nm derived from the Aethalometer measurements during (a) the TROPOS (winter) campaign and (b) the Waldstein (summer) campaign. The fractions of black carbon (dark grey area) and brown carbon (light grey area) to the total measured $b_{abs}$ were calculated as described in Eq. 2.





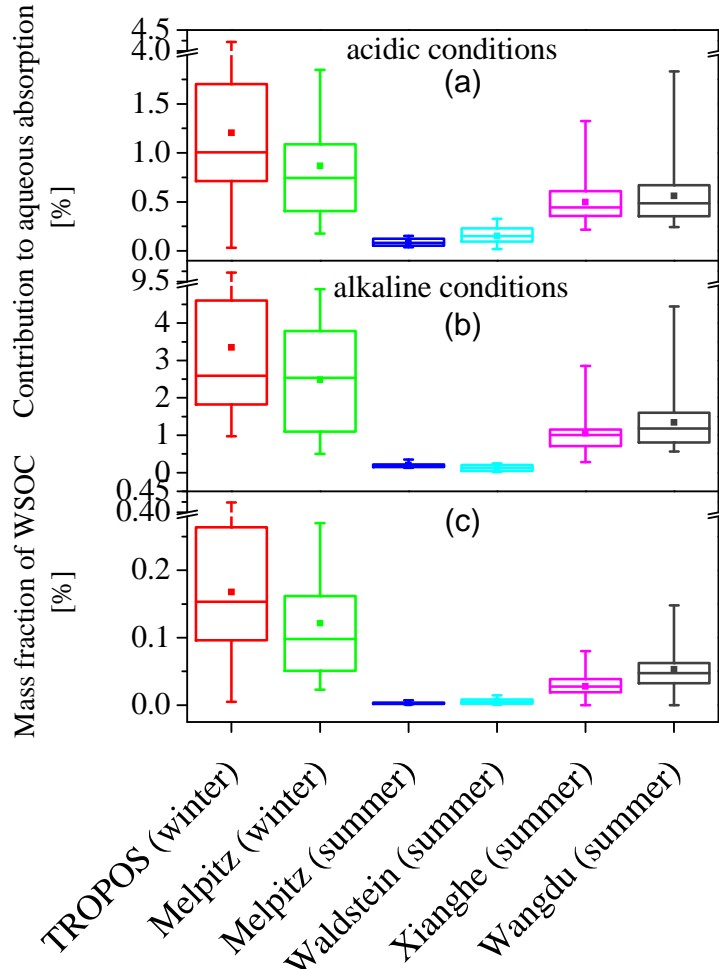

**Figure 4.** Top and center: Contribution of the sum of NACs to the aqueous extract light absorption at each site under acidic (a) and alkaline (b) conditions. Bottom: Mass contributions of the sum of NACs to the WSOC (c).





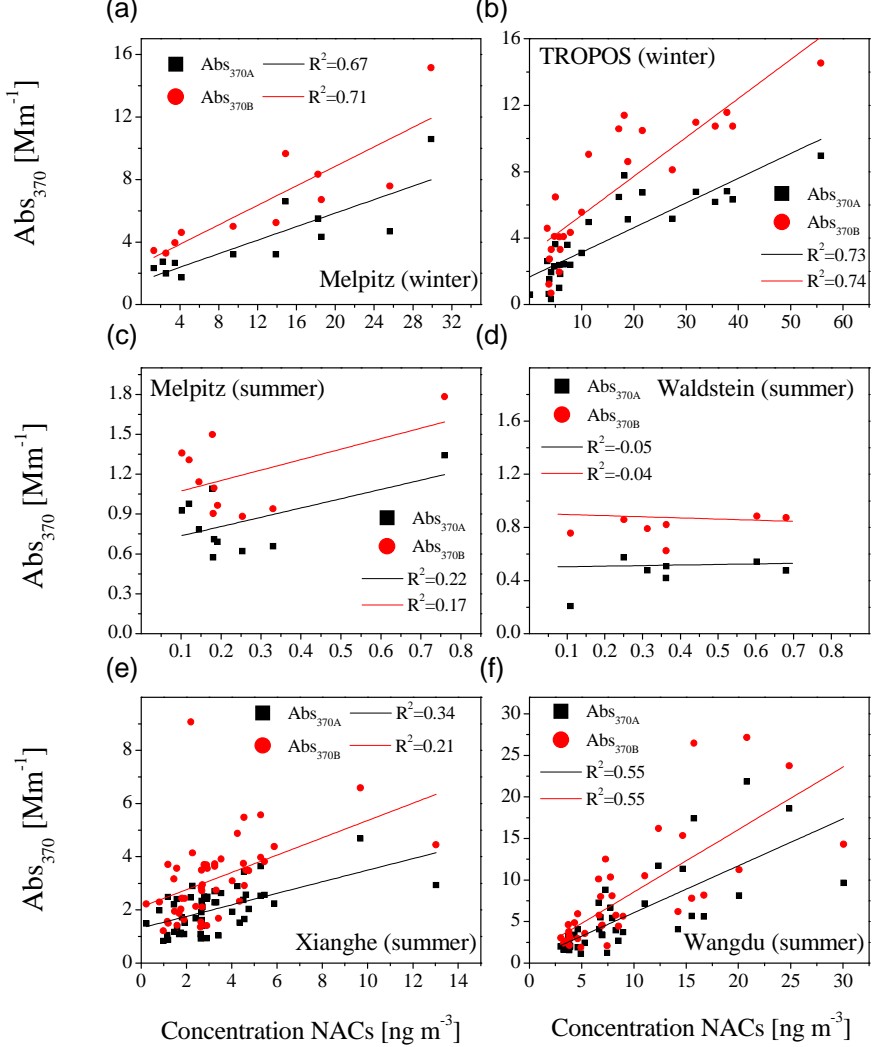

**Figure 5.** Scatter plots of the sum of NACs concentrations with the aqueous light absorption coefficient $Abs_{370}$ for each campaign: (a) Melpitz (winter), (b) TROPOS (winter,), (c) Melpitz (summer), (d) Waldstein (summer), (e) Xianghe (summer), (f) Wangdu (summer). The aqueous light absorption coefficient is given at 370 nm for acidic conditions (indicated by the subscript "A", black squares) and for alkaline conditions (indicated by the subscript "B", red dots).