# Peer review of "Contributions of nitrated aromatic compounds to the light absorption of water-soluble and particulate brown carbon in different atmospheric environments in Germany and China"

_Atmospheric Chemistry and Physics, 2016_

## Referee Comment (RC1) · Anonymous Referee #1 · 25 Sep 2016

Manuscript doi:10.5194/acp-2016-647, 2016

This paper is investigating the concentrations of nitrated aromatic compounds (NACs) in aerosols collected in Germany and China during winter and summer seasons. The contribution of these compounds to the light absorption of BrC PM and water extracts was determined and discussed. This paper is scientifically very interesting and important for understanding the contribution of individual BrC compounds to the light absorption properties of atmospheric aerosols. The manuscript is well written and I have only a few major and minor comments.

[Figure]

Major comment 1. UV-Vis spectra of WSOC fractions were recorded from 300 to 800 nm, however, only 370 nm wavelength was used in the present study (to compare the data with Aethalometer measurements). Different aromatic compounds can have different spectra (see Pretsch et al. 2008, Samburova et al. 2016) and thus their contribution to the BrC absorptivity can vary based on wavelength. It would be very interesting to see how eight NACs contribute to the light absorption of aqueous filter extracts over the spectrum range between 300 and 800 nm.

Major comment 2. Did the author compare the absorption properties of PM water extracts at "natural" pH with artificially acidified and alkylated samples?

Minor comments: Abstract, Line 13. Use (NACs: nitrophenols and nitrated salicylic acid) P. 2, Line 26. "in the UV" means in the "UV-Vis range of the spectrum"? P.2, Line 35. Use "BC" instead of "black carbon", since this abbreviation was introduced above (line 18) P. 3, Lines 6-7. References are needed after "...absorption at 365 nm is used to characterize water-soluble BrC." P. 3, Line 33. Use "biomass-burning aerosols" P. 3, Lines 41-43. This long sentence is a little confusing. P4, Line 9. Add the period (not "until 31 January") for "the first half of the campaign". P4, Line 41. It sounds a little better, but optional: "Analysis of WSOC, levoglucosan, NACs and UV/Vis spectrophotometry measurements were carried out using aqueous filter extracts of different portions of filter in ultrapure water." P. 10, paragraph 1. Why significant BB absorption was observed at the German sites during winter campaigns (because there are more domestic biomass-burning events)? P. 10, Line 23. Use abbreviation "NPs" P. 10, Line 40. Use abbreviation "NACs" P.10, Lines 35-39. If BB was not the major contributor to the analyzed BrC particles, what was a possible source of BrC compounds? Suggestions? P.11, Line 41. Delete space between 4 and %

Summary: I recommend this manuscript for publication after minor revisions.
* * *

---

## Referee Comment (RC2) · Anonymous Referee #2 · 28 Sep 2016

Summary and Overall Recommendation:

This study examines the contribution of nitrated aromatic compounds (NACs) to light absorption of aqueous particle extracts and particulate brown carbon (BrC) from samples collected mainly from Germany and some from China. Aerosol samples were collected onto quartz fiber filters using high-volume filter samplers. The authors focus on 8 NACs, which included nitrophenols and nitrated salicylic acids), previously recognized from prior lab and field studies to contribute to BrC. The novelty in this study lies in the fact that the authors compare the contributions of NACs to WSOC mass and

aqueous aerosol extract light absorption from 4 locations in Germany and 2 locations China. Mainly summer versus winter comparisons are made in Germany and only summer data are obtained from China. The spatial comparisons are quite interesting as the authors know when some of these sites are directly affected by biomass burning (BB events). Overall, this study will be of interest to many readers of ACP; however, there are a lot of technical issues that need to be addressed (as outlined below) before publication can be fully considered. In addition, it would have been more interesting if the authors would have gone further in the chemical characterization of the BrC components collected from these different locations and seasons, especially considering that the NACs didn't explain a larger fraction of the BrC mass. As the authors indicate in the last lines of their paper within the conclusions section, the exciting new results lie in identifying new tracers for BrC that indicate source and chemical process. With that said, I do think many researchers working in BrC aerosol will find this paper interesting due to the use of known BrC constituents (NACs) and comparing their trends between locations and seasons to gain insights into their potential sources. The comparison of the German winter sites to the Chinese summer sites isn't surprising, but it is compelling to see that BB likely contributed to the NACs concentrations at the German winter sites where as other sources (one I mention below in the specific comments section) contribute to NACs levels in China during summer.

Specific Comments:

1.) Important details missing for the high-volume filter sampling protocols:

How were the quartz fiber filters treated before sampling? Were they pre-combusted before sampling, and if so, at what temperatures and for how long? How were these filters stored after collection? Were they stored in pre-combusted Al foil packets or some other kind of container? How long were filters stored before chemical analyses and how did this affect the data presented here? Did the authors determine the recovery efficiencies of NACs from this filter media and was this considered into the calculations for their mass concentrations? The authors stated that the samples were stored at -20

C. Please clarify that this was under dark conditions too of course. Even though it may seem trivial, these details really should be added to the experimental section.

2.) Extraction solvent:

Can the authors comment on how well water extractions remove BrC constituents from the filters? Why wasn't another solvent, such as an organic solvent, considered as well in this study? I ask this question since HULIS-like species, which are likely oligomeric in nature, may not have been well removed from the filter media. As the authors know, HULIS-like species can contribute to the BrC fraction. Lin et al. (2014, ES&T) found that the BrC fraction within IEPOX-derived SOA was highly oligomeric in nature but also less water soluble, so extracting the filters with an organic solvent was really important in discovering these light-absorbing oligomers. This study isn't the only one to consider this issue, but certainly a recent example to consider in terms of extraction solvent.

3.) Levoglucosan:

Since levoglucosan was quantified using IC coupled to PAD, how confident are the authors that there are no co-eluting species? I ask this question since GC/MS with prior derivatization tends to take this concern away due to its high chromatographic resolution.

4.) Changing the pH of aqueous extracts:

By intentionally making extracts acidic or basic, do the authors fear changes in the chemical composition could occur due to unwanted reactions? This is important to think about, especially if one is concerned about the presence of oligomeric species that could degrade via dehydration reactions or other types of unforeseen reactions. I think the authors need to comment on this potential issue. As an example, how might this affect the UV-Vis measurements? I can see this step you have introduced here being confusing to some of the readership of ACP.

5.) BB not the possible source of NACs at the Chinese sites:

Were these NACs during summer in China associated with the photochemical oxidation of anthrpogenic VOCs, such as aromatics? Previous work, such as by the EPA group (Jaoui et al., studies) and Sato et al. (JPCA, 2008), have shown that the photochemical oxidation of aromatic VOCs in the presence of NOx yields NACs. If you collect filters from these experiments, they are brown. So it would be interesting to know if this is correlated with photochemical processing of VOCs (like aromatics) associated with traffic emissions.

6.) Page 10, Line 26:

The authors state "nighttime concentrations were found to be slightly higher than during the day." For statements like this one and elsewhere in the manuscript, is this statistically significant?

7.) Page 11, Line 11:

The authors state "The contribution of NACs to Abs(370 nm) was low for the campaigns Waldstein (summer) and Melpitz (summer)."

Probably not unexpected, right, especially since there is no BB influence or traffic influence? But are there other types of BrC constituents missing, such as those observed from monoterpenes in lab studies by the Laskin and Nizorodov groups? It would be interesting to know what is contributing to the small BrC levels..

Minor Comments:

1.) Fix the numbering of subsections in Section 2.

---

## Referee Comment (RC3) · Anonymous Referee #3 · 29 Sep 2016

General comments

The authors present quantitative data on concentrations and light absorption contributions of eight nitrated aromatic compounds (NACs) measured in atmospheric particles at five different locations in Germany and China during two different seasons.

Light absorption by brown carbon is an important topic for the overall assessment of the direct aerosol effect; many open questions remain related to the extent and organic compounds involved. The diversity of measurement sites and the comparison of two different methods for light absorption assessment make this study very interesting. The

manuscript is well written. I thus recommend publication after the comments below have been addressed.

Specific comments

The focus is on WSOC and water-soluble BrC. What I am missing is an assessment/estimate of the fractions WSOC/OC and water-soluble BrC/BrC, to get an idea of comparability and validity of methods.

I am also missing a direct comparison of babs, the light absorption coefficient of particles, and Abs, and as well the calculated MAE (why was MAE not calculated for the Aethalometer data, based on babs and total PM mass/total PM organic mass, if available?) At least a comparison plot of the relative temporal evolution of these parameters for the Waldstein (summer) and TROPOS (winter) campaigns is highly interesting from both a scientific and methodological point of view and should be added to the paper (could also be in the SI).

P. 2, l. 14 – 15: With the attention brown carbon is getting in recent years (and in the rest of your introduction) this statement seems too strong here. Add "in global climate models" for specification.

P. 3, l. 3: To my knowledge, Sandradewi et al. (Environ. Sci. Technol., 2008, 42 (9), pp 3316–3323, DOI: 10.1021/es702253m) were among the first to introduce the "Aethalometer model" for the separation of BC and BrC (then traffic vs wood burning contributions). Please cite.

P. 4, l. 9 (compare comment for Table 1 and p. 4, l. 38 - 40): Give time interval for 12h mean.

P. 4, l. 33: How were they determined, based on what criteria? Please add this information.

P. 10, l. 27 – 30: Photolysis can be sink of NP as well.

[Figure]

Technical comments:

The abstract is relatively long and dense. It would profit from a bit of streamlining. Consider moving the sentence on p. 1, l. 34 – 24, to l. 34.

P. 2, l. 14: [. . .] are usually treated [. . .]

P. 5, l. 7: What do you mean by "distributed sources"? Not clear.

Table 1: The alignment of some of the columns is off. Please correct for better readability. Also add the sampling times (now give in Table S2) to Table 1.

Table 2: I suggest highlighting the highest and lowest values in each column/category. The light absorption contribution (in %) is given for NP and NSA individually, but there is not further mentioning of this. I assume NP is the sum of the 6 NP and NSA the sum of the 2 NSA you mention on p. 4, l. 33 -35. This kind of differentiation/grouping is only done in Table 2 – I suggest making that consistent throughout the manuscript. Do A) and B) refer to acidic and alkaline conditions? Please clarify and add this information in the table caption.

---

## Author Comment (AC1) · 24 Nov 2016

**Response to Anonymous Referee #1**

This paper is investigating the concentrations of nitrated aromatic compounds (NACs) in aerosols collected in Germany and China during winter and summer seasons. The contribution of these compounds to the light absorption of BrC PM and water extracts was determined and discussed. This paper is scientifically very interesting and important for understanding the contribution of individual BrC compounds to the light absorption properties of atmospheric aerosols. The manuscript is well written and I have only a few major and minor comments.

*The authors would like to thank the reviewer for recommending the paper and the helpful comments. The revision was carried out carefully according to the reviewer's suggestions.*

**1.** Major comment 1. UV-Vis spectra of WSOC fractions were recorded from 300 to 800 nm, however, only 370 nm wavelength was used in the present study (to compare the data with Aethalometer measurements). Different aromatic compounds can have different spectra (see Pretsch et al. 2008, Samburova et al. 2016) and thus their contribution to the BrC absorptivity can vary based on wavelength. It would be very interesting to see how eight NACs contribute to the light absorption of aqueous filter extracts over the spectrum range between 300 and 800 nm.

*Thank you very much for this comment. We agree, that the contribution of NACs to the aqueous extract light absorption over the spectrum range of 300 to 800 nm might be interesting to the reader. Therefore, according to the suggestion of the referee, we added a new diagram to the main script showing the contribution of NACs to Abs over the spectral range of 300 to 500 nm (acidic conditions). This range was chosen, since above 500 nm NACs are only weakly absorbing. A short description of the diagram was added to the text in Section 3.3. Furthermore, a diagram for the contribution of NACs to Abs over 300 to 500 nm under alkaline conditions, as well as the spectral evolution of Abs under acidic and alkaline conditions was added to the supplement.*

*Detailed changes to the manuscript:*
*Additional text added to Section 3.3 (lines and pages are given according to the submitted ACPD manuscript)*

*p. 11, l. 7:*
*"The focus is on the relative contribution to the aqueous extract light absorption at the wavelength of 370 nm to match the 370 nm channel of the Aethalometer. To give additional information, the relative contribution of NACs to Abs (acidic conditions) over a spectral range of 300 to 500 nm is presented in Fig. 5 as their campaign averages."*

*p. 11, l. 18:*
*"The relative contribution of NACs to Abs is wavelength dependent. Under acidic conditions, the contribution of NACs to the aqueous extract light absorption increases towards lower wavelength, reaching a maximum that was found to generally lie in the range of 330 - 350 nm (see Fig. 5). Moreover, it can be seen that maximum values differ for individual compounds. Under alkaline conditions, this maximum shifts towards the range of 400 - 420 nm (see Fig. S6). As stated above, the results for acidic conditions are likely to be more atmospherically relevant. Therefore, a higher influence of NACs towards shorter wavelengths suggests that they can be more important in terms of their influence on atmospheric photochemistry, e.g. $O_3$ photolysis (Jacobson, 1999)."*

*removed sentence:*
*p. 12, l. 10-11:*
*"This suggests that the influence of NACs is higher towards shorter wavelength, i.e. near their absorption maximum and thus can be more important in terms of their influence on atmospheric photochemistry, e.g. O3 photolysis (Jacobson, 1999)."*

*Additional diagram in the manuscript:*

[Figure]

**Figure 5.** *Relative contribution of NACs to Abs$_\lambda$ over the spectral range of 300 to 500 nm for each measurement campaign (a-f) under acidic conditions. The data is presented as campaign averages. Due to instrumental issues, data for lower wavelengths is not always available.*

*Figure numbers of diagram appearing later in the text were changed accordingly.*

*Additional diagrams in the supplement:*

[Figure]

**Figure S6.** *Relative contribution of NACs to Abs$_\lambda$ over the spectral range of 300 to 500 nm for each measurement campaign (a-f) under alkaline conditions. The data is presented as campaign averages. Due to instrumental issues, data for lower wavelengths is not always available.*

Acidic conditions

[Figure]

***Figure S7.*** *Abs$_\lambda$ over the spectral range of 300 to 500 nm for each measurement campaign (a-f) under acidic conditions. The data is presented as campaign averages. Due to instrumental issues, data for lower wavelengths is not always available.*

[Figure]

**Figure S8.** *Abs$_\lambda$ over the spectral range of 300 to 500 nm for each measurement campaign (a-f) under alkaline conditions. The data is presented as campaign averages. Due to instrumental issues, data for lower wavelengths is not always available.*

**2.** Major comment 2. Did the author compare the absorption properties of PM water extracts at "natural" pH with artificially acidified and alkylated samples?

*The pH of the pure water extracts depends on the amount of water used in the extraction process. Furthermore, it would contain a mixture of protonated and deprotonated NACs. To determine the relative contribution of NACs to Abs one would need to know the amount of protonated and deprotonated species in the solution. Under the chosen acidic conditions the NACs are fully protonated and under alkaline conditions fully deprotonated. Thus we determined the lower and upper limits for the contribution of NACs to Abs. Due to the limited amount of aqueous filter exracts (and filter material) we chose to analyse acidified and alkalized filter extracts only.*

**3.** Minor comments: Abstract, Line 13. Use (NACs: nitrophenols and nitrated salicylic acid)
P. 2, Line 26. "in the UV" means in the "UV-Vis range of the spectrum"?

*Here, we do refer to the "UV range of the spectrum" and not to the "UV-Vis range of the spectrum, since the observed reduction of photolysis rates and atmospheric oxidents was reported to be in the UV and not UV-Vis. However, to clarify this sentence "range of the spectrum" was added.*

**4.** P.2, Line 35. Use "BC" instead of "black carbon", since this abbreviation was introduced above (line 18)
P. 10, Line 23. Use abbreviation "NPs" P. 10, Line 40. Use abbreviation "NACs"

*Changes were made according to the referee's suggestion throughout the manuscript.*

**5.** P. 3, Lines 6-7. References are needed after ". . .absorption at 365 nm is used to characterize water-soluble BrC."

*References added "(e.g., Bosch et al. 2014; Cheng et al. 2011; Hecobian et al. 2010)" (These references were already listed in the reference list in the manuscript)*

**6.** P. 3, Line 33. Use "biomass-burning aerosols"

*done*

**7.** P. 3, Lines 41-43. This long sentence is a little confusing.

*To clarify this part, the sentences was divided into two sentences: "The present study aims to expand the understanding of BrC and water-soluble BrC by investigating its spatial and temporal variation in different, very diverse environments (urban, rural, biogenic, high BB influence). Furthermore, the contributions of individual light-absorbing organic compounds to the BrC light absorption is included."*

**8.** P4, Line 9. Add the period (not "until 31 January") for "the first half of the campaign".

*replaced with "(24 to 31 January)"*

**9.** P4, Line 41. It sounds a little better, but optional: "Analysis of WSOC, levoglucosan, NACs and UV/Vis spectrophotometry measurements were carried out using aqueous filter extracts of different portions of filter in ultrapure water."

*This sentence was adopted to the manuscript according to the referee's suggestion*

**10.** P. 10, paragraph 1. Why significant BB absorption was observed at the German sites during winter campaigns (because there are more domestic biomass-burning events)?

*As stated in Section 2.1, we concluded that BB was playing a major role at the German winter sites because of high observed levoglucosan concentrations. It is also known that BB due to domestic heating can play a large role in this area.*

**11.** P.10, Lines 35-39. If BB was not the major contributor to the analyzed BrC particles, what was a possible source of BrC compounds? Suggestions?

*A similar question was asked by the anonymous Referee#2. Based on the comment by the anonymous Referee#2 and anonymous Referee#1 a possible source for NACs was added to Section 3.2.*

*Added text:*
*p.10, l. 39:*
*"It was found that NACs can be a product of the photochemical processing of anthropogenic volatile organic compounds (Jaoui et al. 2008), which might be a possible source for NACs at the Chinese sites besides BB." was added to the text.*

*Additional reference:*
*"Jaoui, M., Edney, E. O., Kleindienst, T. E., Lewandowski, M., Offenberg, J. H., Surratt, J. D., and Seinfeld, J. H.: Formation of secondary organic aerosol from irradiated α-pinene/toluene/NOx mixtures and the effect of isoprene and sulfur dioxide, J. Geophys. Res., 113, D09303, doi: 10.1029/2007JD009426, 2008."*

**12.** P.11, Line 41. Delete space between 4 and % Summary: I recommend this manuscript for publication after minor revisions.

*Already published papers in ACP do use a space between the number and %. Therefore, we chose to also use a space between the number and % and the corrections were made accordingly throughout the text.*

---

## Author Comment (AC2) · 24 Nov 2016

**Response to Anonymous Referee #2**

Summary and Overall Recommendation:
This study examines the contribution of nitrated aromatic compounds (NACs) to light absorption of aqueous particle extracts and particulate brown carbon (BrC) from samples collected mainly from Germany and some from China. Aerosol samples were collected onto quartz fiber filters using high-volume filter samplers. The authors focus on 8 NACs, which included nitrophenols and nitrated salicylic acids), previously recognized from prior lab and field studies to contribute to BrC. The novelty in this study lies in the fact that the authors compare the contributions of NACs to WSOC mass and aqueous aerosol extract light absorption from 4 locations in Germany and 2 locations China. Mainly summer versus winter comparisons are made in Germany and only summer data are obtained from China. The spatial comparisons are quite interesting as the authors know when some of these sites are directly affected by biomass burning (BB events). Overall, this study will be of interest to many readers of ACP; however, there are a lot of technical issues that need to be addressed (as outlined below) before publication can be fully considered. In addition, it would have been more interesting if the authors would have gone further in the chemical characterization of the BrC components collected from these different locations and seasons, especially considering that the NACs didn't explain a larger fraction of the BrC mass. As the authors indicate in the last lines of their paper within the conclusions section, the exciting new results lie in identifying new tracers for BrC that indicate source and chemical process. With that said, I do think many researchers working in BrC aerosol will find this paper interesting due to the use of known BrC constituents (NACs) and comparing their trends between locations and seasons to gain insights into their potential sources. The comparison of the German winter sites to the Chinese summer sites isn't surprising, but it is compelling to see that BB likely contributed to the NACs concentrations at the German winter sites where as other sources (one I mention below in the specific comments section) contribute to NACs levels in China during summer.

*We would like to thank the referee for the recommendation and the helpful suggestions. We agree, that it is indeed interesting to characterize BrC components further. Therefore, we are going to consider also other possible BrC compounds in forthcoming studies. Nevertheless, the focus of this study was to characterize NACs as BrC components.*

**Specific Comments**:
**1.** Important details missing for the high-volume filter sampling protocols: How were the quartz fiber filters treated before sampling? Were they pre-combusted before sampling, and if so, at what temperatures and for how long? How were these filters stored after collection? Were they stored in pre-combusted Al foil packets or some other kind of container? How long were filters stored before chemical analyses and how did this affect the data presented here? Did the authors determine the recovery efficiencies of NACs from this filter media and was this considered into the calculations for their mass concentrations? The authors stated that the samples were stored at -20 °C. Please clarify that this was under dark conditions too of course. Even though it may seem trivial, these details really should be added to the experimental section.

*Details on the high-volume filter sampling protocol were added to Section 2.2 as suggested by the referee.*
*The paragraph reads now like this (p. 4, l. 37 – 40):*
*"PM$_{10}$ was collected on quartz fiber filters with a Digitel DHA-80 high volume filter sampler (MK 360, Munktell, Falun, Sweden, flow rate: 0.5 m3 min-1). To minimize blank content, the filters were pre-baked for 24 h at 105 °C. Day and night samples (11 h or 12 h, see Table 1)*

*were taken during each campaign except for Melpitz (winter) and Melpitz (summer), where particles were collected for 24 h. After sampling, filters were stored in clean aluminium tins at -20 °C in the dark until extraction (extraction was done within a year after sampling). It is assumed, that storage at -20 °C prevents chemical degradation of the sample."*

*Regarding the question about recovery efficiencies from filter media:*
*An accurate determination of recovery efficiencies from filter media is not easily possible. One might spike a blank filter with a standard solution to determine recoveries from the filter. However, since the target compounds are actually part of an aerosol particle and extracted from aerosol particles and not the filter media, this method is not accurate. The investigated NAC compounds in our study are water soluble. Hence, due to the relatively large amount of water used in the extraction process and the water solubility of NACs, it is expected that NACs dissolve completely into water.*

*Referring to recovery efficiencies of the used enrichment method: Matrix effects and recovery efficiencies in the HF-LPME method are considered in the calculation of nitrophenol mass concentration according to the published method in Teich et al. 2014.*

**2.** Extraction solvent:
Can the authors comment on how well water extractions remove BrC constituents from the filters? Why wasn't another solvent, such as an organic solvent, considered as well in this study? I ask this question since HULIS-like species, which are likely oligomeric in nature, may not have been well removed from the filter media. As the authors know, HULIS-like species can contribute to the BrC fraction. Lin et al. (2014, ES&T) found that the BrC fraction within IEPOX-derived SOA was highly oligomeric in nature but also less water soluble, so extracting the filters with an organic solvent was really important in discovering these light-absorbing oligomers. This tudy isn't the only one to consider this issue, but certainly a recent example to consider in terms of extraction solvent.

*We agree, that methanol or other organic solvents may remove BrC constituents with a higher efficiency from filters than water. However, the solvent of choice is also dependent on the target compounds and the overall aims of the study. In our case, the focus of our study was on NACs. NACs are expected to be fully soluble in water. Considering that water is also a natural solvent in the atmosphere, we believe that water as a solvent was an appropriate choice for our study. In consequence, we also focused on the water-solube BrC light absorption. To assess the contribution to the total BrC light absorption, Aethalometer measurements were included into the study.*
*For forthcoming studies, however, organic solvents are also considered to include less water-soluble BrC constituents, like the mentioned oligomers.*

**3.** Levoglucosan:
Since levoglucosan was quantified using IC coupled to PAD, how confident are the authors that there are no co-eluting species? I ask this question since GC/MS with prior derivatization tends to take this concern away due to its high chromatographic resolution.

*The method used in this study to determine levoglucosan has been published as Iinuma et al. 2009. In this publication, it was stated that the method enables the separation of levoglucosan and arabitol, which was an issue in previous studies. Furthermore, an intercomparison study by Yttri et al. 2015 (Atmos. Meas. Tech.) showed that the used method (high-performance anion-exchange chromatography (HPAEC) with pulsed amperometric detection (PAD))*

*delivers comparable results as methods using GC/MS. Therefore, we are confident that this data is reliable.*

**4.** Changing the pH of aqueous extracts:
By intentionally making extracts acidic or basic, do the authors fear changes in the chemical composition could occur due to unwanted reactions? This is important to think about, especially if one is concerned about the presence of oligomeric species that could degrade via dehydration reactions or other types of unforeseen reactions. I think the authors need to comment on this potential issue. As an example, how might this affect the UV-Vis measurements? I can see this step you have introduced here being confusing to some of the readership of ACP.

*This step was introduced to investigate the NACs and their contribution to the light absorption in an environment where they are either fully protonated or deprotonated. Without this step, a mixture of protonated and deprotonated forms would have been present in the solution. By introducing a change in pH, the upper and lower limits for the contribution of NACs to the BrC light absorption could be determined. To minimize the risk of potential chemical modification the solutions where kept in the dark and analyzed as soon as possible after preparation by UV-Vis-spectrophotometry.*
*For more clarity, an additional text was added to the Section 2.4 in the manuscript (p. 6, l. 4):*

*"to obtain the lower and upper limit for the contribution of NACs to the BrC light absorption. In principle, it could be possible that introducing acids or bases into the system induces unforeseen chemical reactions influencing the total light absorption of the aqueous extract. However, To minimize the risk of potential chemical modification the solutions where kept in the dark and analyzed as soon as possible after preparation by UV-Vis-spectrophotometry."*

**5.** BB not the possible source of NACs at the Chinese sites:

Were these NACs during summer in China associated with the photochemical oxidation of anthropogenic VOCs, such as aromatics? Previous work, such as by the EPA group (Jaoui et al., studies) and Sato et al. (JPCA, 2008), have shown that the photochemical oxidation of aromatic VOCs in the presence of NOx yields NACs. If you collect filters from these experiments, they are brown. So it would be interesting to know if this is correlated with photochemical processing of VOCs (like aromatics) associated with traffic emissions.

*Thank you very much for this comment. From our field experiments alone, it is difficult to attribute concrete sources to the observed NAC concentrations. Unfortunately, there is no VOC data available. Mentioned possible sources in the manuscript are therefore very speculative. Nevertheless, NACs may derive from photochemical processing of VOCs transported to the site. Hence, according to the referee's suggestion this possible source was added to Section 3.2.*
*"It was found that NACs can be a product of the photochemical processing of anthropogenic volatile organic compounds (Jaoui et al. 2008), which might be a possible source for NACs at the Chinese sites besides BB." was added to the text. (p.10, l. 39)*

*Additional reference:*
*"Jaoui, M., Edney, E. O., Kleindienst, T. E., Lewandowski, M., Offenberg, J. H., Surratt, J. D., and Seinfeld, J. H.: Formation of secondary organic aerosol from irradiated α-pinene/toluene/NOx mixtures and the effect of isoprene and sulfur dioxide, J. Geophys. Res., 113, D09303, doi: 10.1029/2007JD009426, 2008."*

**6.** Page 10, Line 26:

The authors state "nighttime concentrations were found to be slightly higher than during the day." For statements like this one and elsewhere in the manuscript, is this statistically significant?

*We checked the data using the t-test method and the slightly higher concentrations observed at nighttime were found to be not statistically significant. To clarify the remark "(not statistically significant at 95 % confidence level)" was added to the text (p. 10, l. 26).*
*The manuscript was checked for similar statements. In other instances in the manuscript, where comparisons where made, clear differences where seen in the data set. In this cases, the information of the statistically significance was not added.*

**7.** Page 11, Line 11:
The authors state "The contribution of NACs to Abs(370 nm) was low for the campaigns Waldstein (summer) and Melpitz (summer)." Probably not unexpected, right, especially since there is no BB influence or traffic influence? But are there other types of BrC constituents missing, such as those observed from monoterpenes in lab studies by the Laskin and Nizorodov groups? It would be interesting to know what is contributing to the small BrC levels.

*We agree to the referee's comment. Since the focus of this study was on NACs, we cannot provide further information of BrC constituents derived from biogenic emission from our data set. However, a possible source for the observed light absorption, mentioned in the literature, was added to Section 3.3.*

*Additional text:*
*p. 11, l. 13:*
*"This result is not surprising, due to the low influence of BB aerosols or traffic and thus low NAC concentrations. A recent study by Nguyen et al. 2013 suggested the formation of BrC from ketoaldehydes derived from biogenic monoterpenes in the presence of ammonium ions. Thus, this reaction may play a role in regions with higher influence of biogenic emissions and might be one explanation for the observed absorption at the Melpitz (summer) and Waldstein (summer) campaigns. The formed species were suggested to consist of conjugated aldol condensates, secondary imines and nitrogen containing heterocycles."*

*Additional reference:*
*"Nguyen, T. B., Laskin, A., Laskin, J., and Nizkorodov, S. A.: Brown carbon formation from ketoaldehydes of biogenic monoterpenes, Faraday Discuss., 165, 291-315, doi: 10.1039/C3FD00036B, 2013."*

**8.** Fix the numbering of subsections in Section 2.
*The numbering was fixed.*

Literature:

Iinuma, Y., Engling, G., Puxbaum, H., and Herrmann, H.: A highly resolved anion-exchange chromatographic method for determination of saccharidic tracers for biomass combustion and primary bio-particles in atmospheric aerosol, Atmos. Environ., 43, 1367-1371, doi: 10.1016/j.atmosenv.2008.11.020, 2009.

Teich, M., van Pinxteren, D., and Herrmann, H.: Determination of nitrophenolic compounds from atmospheric particles using hollow-fiber liquid-phase microextraction and capillary electrophoresis/mass spectrometry analysis, Electrophoresis, 35, 1353-1361, doi: 10.1002/elps.201300448, 2014.

Yttri, K. E., Schnelle-Kreis, J., Maenhaut, W., Abbaszade, G., Alves, C., Bjerke, A., Bonnier, N., Bossi, R., Claeys, M., Dye, C., Evtyugina, M., García-Gacio, D., Hillamo, R., Hoffer, A., Hyder, M., Iinuma, Y., Jaffrezo, J.-L., Kasper-Giebl, A., Kiss, G., López-Mahia, P. L., Pio, C., Piot, C., Ramirez-Santa-Cruz, C., Sciare, J., Teinilä, K., Vermeylen, R., Vicente, A., and Zimmermann, R.: An intercomparison study of analytical methods used for quantification of levoglucosan in ambient aerosol filter samples, Atmos. Meas. Tech., 8, 125-147, doi:10.5194/amt-8-125-2015, 2015.

---

## Author Comment (AC3) · 24 Nov 2016

**General comments**
The authors present quantitative data on concentrations and light absorption contributions of eight nitrated aromatic compounds (NACs) measured in atmospheric particles at five different locations in Germany and China during two different seasons. Light absorption by brown carbon is an important topic for the overall assessment of the direct aerosol effect; many open questions remain related to the extent and organic compounds involved. The diversity of measurement sites and the comparison of two different methods for light absorption assessment make this study very interesting. The manuscript is well written. I thus recommend publication after the comments below have been addressed.

*We would like to thank the referee for the recommendation and the helpful suggestions*

**Specific comments**
**1.** The focus is on WSOC and water-soluble BrC. What I am missing is an assessment/estimate of the fractions WSOC/OC and water-soluble BrC/BrC, to get an idea of comparability and validity of methods. I am also missing a direct comparison of babs, the light absorption coefficient of particles, and Abs, and as well the calculated MAE (why was MAE not calculated for the Aethalometer data, based on babs and total PM mass/total PM organic mass, if available?) At least a comparison plot of the relative temporal evolution of these parameters for the Waldstein (summer) and TROPOS (winter) campaigns is highly interesting from both a scientific and methodological point of view and should be added to the paper (could also be in the SI).

*We agree, that this information might be interesting to the reader. Therefore, as suggested by the referee, a diagram was added to the supplement containing the temporal variation of $MAE_{370}$ for particulate BrC, the fraction WSOC/OC and the fraction of aqueous extract light absorption to particulate BrC light absorption for the campaigns TROPOS (winter) and Waldstein (summer). Abs and $b_{abs}$ are not directly comparable. According to the method by Liu et al. 2013 (Atmos. Chem. Phys) a conversion factor of 2 was applied to Abs. A brief comparison of Abs and $b_{abs}$ is given in Section 3.1.3.*

*Additional text:*
*p. 9, l. 2:*
*"Normalizing the determined babs for particulate BrC by the according OC content gives the mass absorption efficiency for BrC in the particle (MAE370, BrC, particle). The temporal variation of MAE370, BrC, particle is displayed in Fig. S4."*

*p. 9, l. 8*
*"and an average $MAE_{370,\ BrC,\ particle}$ of 1.95 $m^2\ g^{-1}$ was calculated"*

*p. 9., l. 19:*
*", a $MAE_{370,\ BrC,\ particle}$ of 0.21 $m^2\ g^{-1}$" and "The temporal variation of the fraction of the converted Abs370 to the particulate BrC light absorption is given in Fig. S4."*

*Additional diagrams in the supplement:*

[Figure]

***Figure S4.*** *Temporal variation of MAE$_{370, BrC, particle}$ for particulate BrC, the WSOC/OC fraction and the fraction of aqueous extract light absorption to particulate BrC light absorption for the campaigns TROPOS (winter) (a-c) and Waldstein (summer) (d-f). For comparability of aqueous extract light absorption and the particulate BrC light absorption, Abs$_{370A}$ (acidic conditions) was multiplied by a factor of 2, according to the method mentioned in the main text. MAE$_{370, BrC, particle}$ was determined by normalizing b$_{abs, 370, BrC}$ by the according OC content of the sample.*

**2.** P. 2, l. 14 – 15: With the attention brown carbon is getting in recent years (and in the rest of your introduction) this statement seems too strong here. Add "in global climate models" for specification.

*Additions were made according to the referee's suggestions.*

*Additional text:*
*p. 2., l. 14:*
*"in global climate models"*

**3.** P. 3, l. 3: To my knowledge, Sandradewi et al. (Environ. Sci. Technol., 2008, 42 (9), pp 3316–3323, DOI: 10.1021/es702253m) were among the first to introduce the "Aethalometer model" for the separation of BC and BrC (then traffic vs wood burning contributions). Please

cite.

*Additions were made according to the referee's suggestions.*

*Additional text:*
*p. 3, l. 3*
*"Sandradewi et al. 2008"*

*Additional reference:*
*"Sandradewi, S., Prevot, A. S. H., Szidat, S., Perron, N., Alfarra, M. A. Lanz, V. A., Weingartner, E., and Baltensperger, U.: Using Aerosol Light Absorption Measurements for the Quantitative Determination of Wood Burning and Traffic Emission Contributions to Particulate Matter, Environ. Sci. Technol., 42, 3316-3323, doi: 10.1021/es702253m, 2008."*

**4.** P. 4, l. 9 (compare comment for Table 1 and p. 4, l. 38 - 40): Give time interval for 12h mean.

*see reply to 10.*

**5.** P. 4, l. 33: How were they determined, based on what criteria? Please add this information.

*The determination of NACs was explained later in the text. To avoid confusion, specific NAC compounds are now mentioned later in the text*

*p. 4, l. 33 - 36 were moved to p. 5, l. 13.*

**6.** P. 10, l. 27 - 30: Photolysis can be sink of NP as well.

*"or photolysis processes" added to p. 10, l. 28.*

Technical comments:

**7.** The abstract is relatively long and dense. It would profit from a bit of streamlining. Consider moving the sentence on p. 1, l. 34 – 24, to l. 34.

*As suggested by the referee, to improve the readability, the abstract was shortened.*

*Additional text:*
*p.1, l. 25*
*"with larger values at higher pH"*

*Removed sentences:*

*p. 1, l. 25 - 26:*
*": at pH 10, the aqueous light absorption coefficient Abs370 and the mass absorption efficiency (MAE370) at 370 nm were a factor of 1.6 and 1.4 larger than at pH 2, respectively"*

*p. 1. l. 29 - 31:*
*"Furthermore, it was found that the $MAE_{370}$ values in 30 winter in Germany exceeded those of the Chinese summer background stations (average of $0.85 \pm 0.24$ $m^2$ $g^{-1}$ compared to $0.47 \pm 0.15$ $m^2$ $g^{-1}$)"*

*p. 1. l. 36 - p. 2. l. 1:*
*"The absorption Ångström exponent of the ambient aerosol during the campaigns at TROPOS (winter) and Waldstein (summer) was found to be 1.5±0.1 and 1.2±0.3, respectively."*

*p. 2, l. 8 - 9:*
*"A correlation of NAC concentrations with $Abs_{370}$ was observed for the BB-influenced campaigns at TROPOS (winter) and Melpitz (winter)."*

**8.** P. 2, l. 14: [. . .] are usually treated [. . .]

*corrected*

**9.** P. 5, l. 7: What do you mean by "distributed sources"? Not clear.

*Here, we meant it in the sense of "various" or "different". For clarity the word "distributed" was replaced by "various"*

**10.** Table 1: The alignment of some of the columns is off. Please correct for better readability. Also add the sampling times (now give in Table S2) to Table 1.

*Table 1 was corrected and the sampling times were added. Table S1, that originally contained the sampling times, was removed from the supplement.*

**11.** Table 2: I suggest highlighting the highest and lowest values in each column/category. The light absorption contribution (in %) is given for NP and NSA individually, but there is not further mentioning of this. I assume NP is the sum of the 6 NP and NSA the sum of the 2 NSA you mention on p. 4, l. 33 -35. This kind of differentiation/grouping is only done in Table 2 – I suggest making that consistent throughout the manuscript. Do A) and B) refer to acidic and alkaline conditions? Please clarify and add this information in the table caption.

*"Acidic conditions are indicated by the letter "A" and alkaline conditions are indicated by the letter "B"" was added to the table caption.*
*As mentioned by the referee, the differentiation for NPs and NSAs was made in Table 2 only. To be consistent with the text, we changed the information in Table 2 to values for NACs instead NPs + NSAs. Furthermore the highest and lowest mean values for each category were marked in bold, as suggested by the referee.*

Literature

Liu, J., Bergin, M., Guo, H., King, L., Kotra, N., Edgerton, E., and Weber, R. J.: Size-resolved measurements of brown carbon in water and methanol extracts and estimates of their contribution to ambient fine-particle light absorption, Atmos. Chem. Phys., 13, 12389-12404, doi: 10.5194/acp-13-12389-2013, 2013.